# Ambiphilic cross-coupling with aryl-bismuth reagents

Byeongdo Roh[1,2], Benedict A. Williams[1] & Josep Cornella[1✉]

Cross-coupling reactions traditionally permit the formation of Ar-Ar bonds between an aryl nucleophile and an aryl electrophile under transition-metal catalysis[1,2]. The high selectivity of the myriad of couplings known to date relies on a tailored combination of nucleophilic and electrophilic coupling partners, enabled by the mechanistic distinction between nucleophiles and electrophiles, which undergo fundamentally different catalytic steps[3]. Here we report ambiphilic aryl-bismuth reagents that can behave as either nucleophiles or electrophiles in transition-metal-catalysed cross-couplings, fundamentally breaking from this dichotomy in reactivity. Their ambiphilic reactivity arises from their ability to engage in both oxidative addition and transmetalation processes with transition-metal complexes, as demonstrated by stoichiometric and mechanistic studies. By demonstrating that a single aryl reagent can engage in both canonical elementary steps, this work challenges the long-standing assumption that intrinsic bond polarity rigidly dictates the mechanistic role in cross-coupling chemistry.

Transition-metal-catalysed cross-coupling reactions have revolutionized carbon–carbon (C–C) bond formation, offering straightforward, selective and modular routes to pharmaceuticals, materials and agrochemicals (Fig. 1a)[1,2]. The cross-selectivity in these reactions is achieved by employing two electronically distinct coupling partners: an aryl nucleophile (Ar-$Y$) and an aryl electrophile (Ar-$X$), which interface with the transition-metal catalyst at different points in the catalytic cycle[3]. Successful aryl nucleophiles (Ar-$Y$) are prefunctionalized aromatic substrates bearing either a carbon-metal bond (C-[M]) ($Y$ = Li, Zn$X$, Mg$X$ and so on) or a carbon-heteroatom bond (C-[MG]) ($Y$ = B(OR)$_2$, SnR$_3$, SiR$_3$ and so on)[4–7], which polarize the C atom with enough $\delta^-$ charge to undergo successful transmetalation with the metal catalyst (Fig. 1a, left). On the other hand, the aryl electrophiles (Ar-$X$) are equipped with a leaving group $X$ ($X$ = Cl, Br, I, OTf, $^+$SR$_2$ and so on), which imparts $\delta^+$ charge to the attached C, enabling oxidative addition of a low-valent transition-metal centre (Fig. 1a, right)[8–13]. Therefore, the mechanistic origin of the exquisite selectivity of these couplings is embedded in the reaction design: aryl nucleophiles do not interfere with the elementary steps performed by the electrophiles and vice versa. That is, although aryl electrophiles exclusively engage with the transition-metal catalyst by means of oxidative addition, the aryl nucleophiles exclusively undergo transmetalation with the metal centre, which thus ensures high cross-selectivity.

Although traditional cross-coupling relies on the complementary reactivity of nucleophiles and electrophiles, recent advancements have expanded the scope of this reaction class to include the coupling of two aryl nucleophiles[14,15] or two aryl electrophiles[16]. Although powerful, these transformations require carefully selected oxidants or reductants, respectively, to fulfil a balance in redox enabling turnover of the catalyst. Despite this progress, the mechanistic framework of cross-coupling (oxidative addition, transmetalation, reductive elimination) remains dictated by the nature of the aryl synthons involved (either Ar-$Y$ or Ar-$X$), often imposing inherent limitations on reaction scope and selectivity.

In this work, we introduce a conceptually distinct approach to cross-coupling, which decouples the synthetic design from the polarity constraints of the coupling partners that have long been a central assumption in the field. Specifically, we show how the unique ambiphilic properties of organobismuth(III) complexes permit their exploitation as both nucleophiles and electrophiles (Fig. 1b). This dual reactivity manifold is the consequence of the intrinsic redox reactivity of the Bi(III) complexes, which undergo either transmetalation or oxidative addition depending on the coupling partner employed. This conceptually distinct cross-coupling strategy expands retrosynthetic flexibility in the synthesis of (hetero)biaryls by allowing a single aryl synthon to be employed as either a nucleophile or an electrophile (Fig. 1b). This enables the selection of coupling partners based on their synthetic accessibility or the opportunity to introduce orthogonally reactive handles for downstream functionalization. Isolable, bench stable aryl-bismuth reagents provide a modular platform for coupling reactions: in diversification of complex or high value aryl fragments they prevent repeated refunctionalization steps in the interconversion between nucleophiles and electrophiles, and thus enable expedient parallel synthesis across both electrophiles and nucleophiles from a single intermediate.

## Discovery of ambiphilic coupling

Aryl-bismuth reagents have long been used as nucleophilic arylating agents[17–22]. Indeed, when aryl-bismuth(III) species **PhBi$^{III}$Br** is used in cross-coupling with **ArI-1** using catalytic Pd(PPh$_3$)$_4$, NaI as an additive and DMA ($N$,$N$-dimethylacetamide) as the solvent, the desired cross-coupled product **1** is obtained in 79% yield, alongside 26% yield of homocoupling side product **2** (Fig. 2a(i)) (see Supplementary Fig. 3

[1]Max-Planck-Institut für Kohlenforschung, Mülheim an der Ruhr, Germany. ✉e-mail: cornella@kofo.mpg.de

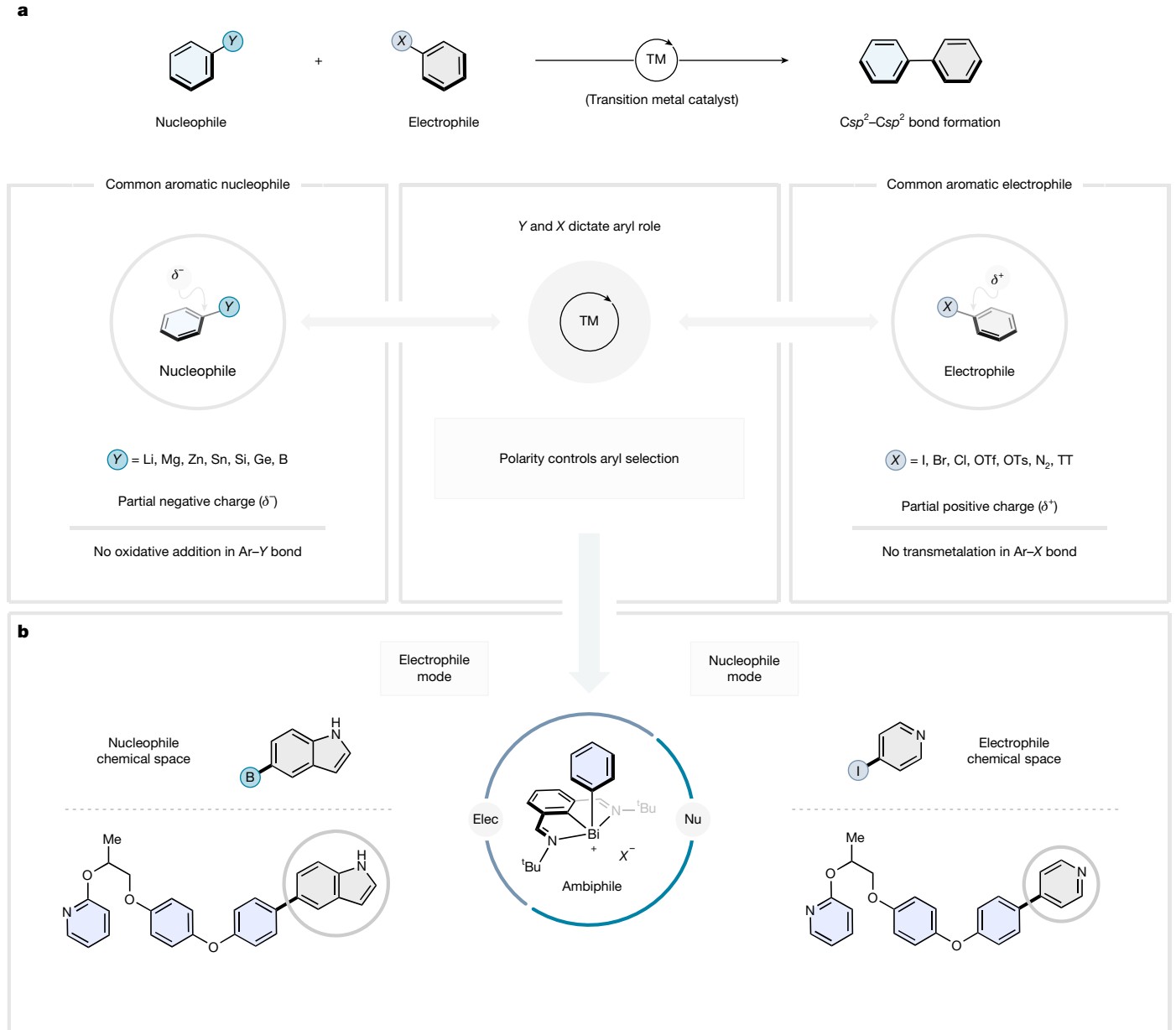

**Fig. 1 | Ambiphilic cross-coupling with organobismuth reagents.**
**a**, Fundamental polarity match between the two coupling reagents in a cross-coupling: aryl nucleophiles couple with aryl electrophiles. **b**, Aryl-bismuth(III) complexes can behave as either aryl electrophiles or aryl nucleophiles in cross-coupling. Elec, electrophile; Nu, nucleophile; TM, transition metal.

for details). Preformed oxidative addition complex (**OAC-1**) could replace Pd(PPh₃)₄ as a precatalyst for the transformation, furnishing cross-coupled product **1** in 84% yield. However, when control experiments using stoichiometric amounts of the **OAC-1** with **PhBiᴵᴵᴵBr** were performed under various conditions, the desired product **1** was consistently obtained in diminished yields (Fig. 2a(ii)). Although these observations validate that aryl-bismuth species undergo transmetalation with Ar–Pd(II) intermediates[23], the lower yields were indicative of side reactivity of the Pd species. Interestingly, addition of aryl iodide (**ArI-1**) restored the reactivity to 80% yield of product **1**. We hypothesized that, in the absence of aryl iodide, the resulting Pd(0) complex can also react with **PhBiᴵᴵᴵBr**. Indeed, Bi–C cleavage with transition metals has been observed[17,24]; however, the difficult oxidative addition to Bi(III)–C bonds and instability of low-valent Bi species that ensue precluded further synthetic exploitation. With this hypothesis, we devised that the *N,C,N*-pincer scaffold could accommodate Bi(III)/Bi(I) redox fluctuations, and thus turn these aryl-bismuth(III) species

into competent electrophiles. To verify this hypothesis, **PhBiᴵᴵᴵBr** was directly combined with Pd(PPh₃)₄, which furnished the Pd(II) oxidative addition complex along with the *N,C,N*-pincer bismuthinidene (**Bi(I)**) (Fig. 2b(i)). Aryl-bismuth complexes bearing non-coordinating counteranions, such as BF₄ or OTf, also reacted in the presence of Pd(PPh₃)₄, affording the corresponding **Bi(I)** species in high yields. However, in these cases, no oxidative addition complexes were observed, attributable to the reported instability of the ensuing cationic Pd(II) intermediates[25–27]. In an attempt to capture such fleeting cationic Ar–Pd(II) intermediates, we performed the reaction with a transmetalating agent, namely, a boronic acid derivative (Fig. 2b(ii)). When various **PhBiᴵᴵᴵX** were combined with stoichiometric Pd(PPh₃)₄ and **ArB-1**, the desired product **1** was obtained in good yields (40–78%). In addition, a characteristic bismuth-ligand coupling side product was occasionally observed, presumably arising from C–C coupling between the aryl group bound to bismuth and the aryl backbone of the *N,C,N*-pincer ligand, further supporting the ambiphilic reactivity of the aryl-bismuth complex

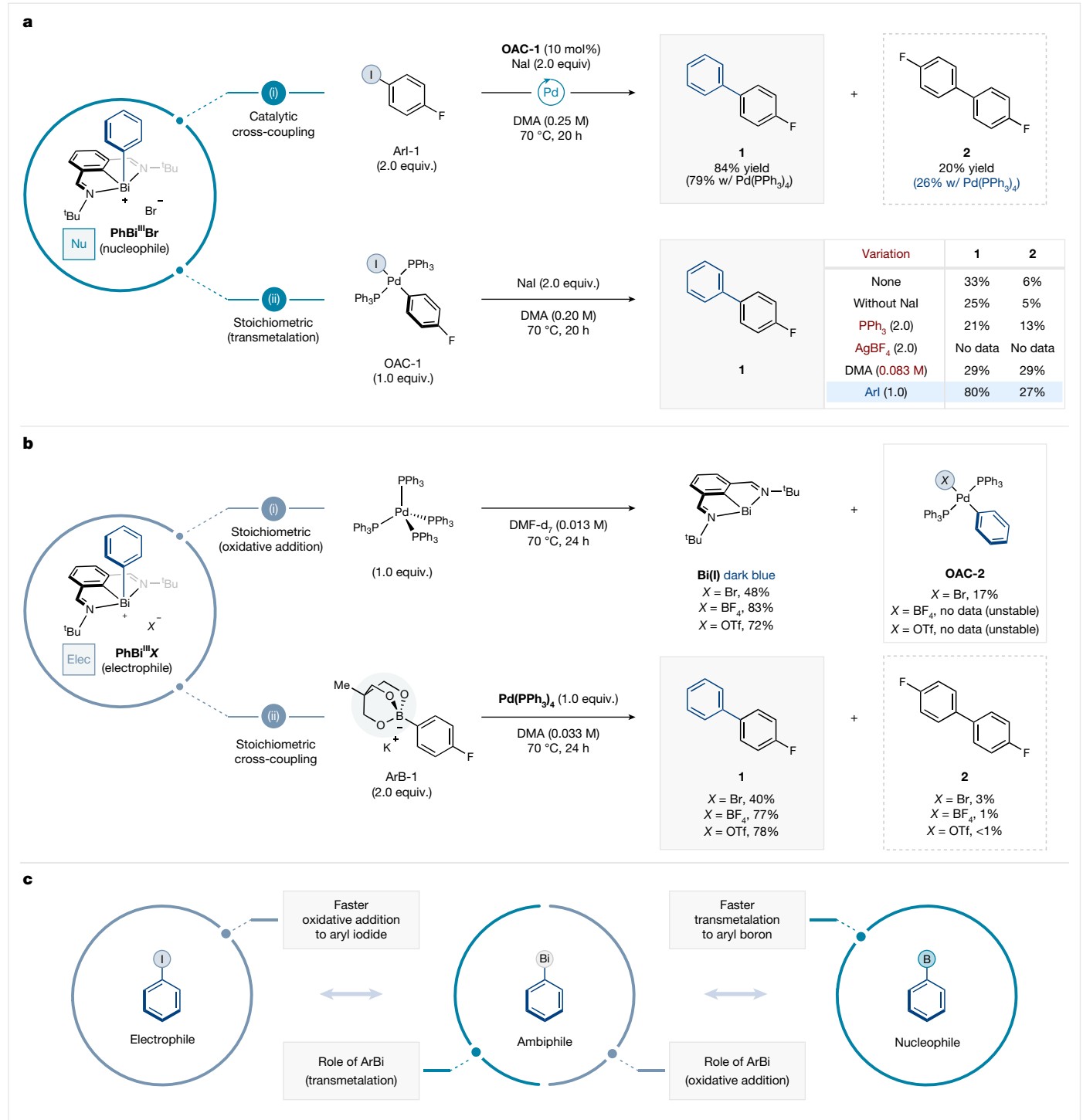

**Fig. 2 | Ambiphilic organobismuth reagents. a**, Aryl-bismuth(III) species effectively transmetalates to Pd(II)–Ar complexes. **b**, Aryl-bismuth(III) species undergoes oxidative addition to Pd(0) complexes. **c**, Mechanistic blueprint strategy for ambiphilic coupling with aryl-bismuth complexes. DMF, *N,N*-dimethylformamide; OAC, oxidative addition complex. Concentrations in parentheses are based on the aryl-bismuth species.

(see Supplementary Fig. 5 for details). Based on these observations, we envisioned a new type of catalytic cross-coupling that departs from the canonical dichotomy of nucleophilic/electrophilic coupling partners as non-overlapping classes of reagents. Instead, an aryl-bismuth compound can behave either as the nucleophile or electrophile depending on the coupling partner employed (Fig. 2c). For example, in a cross-coupling with aryl iodides, the rate of oxidative addition of low-valent Pd into aryl iodides outcompetes oxidative addition into the aryl-bismuth compound, selectively generating the oxidative addition

complex derived from the aryl iodide and the Pd(0) complex. At this stage, the aryl-bismuth species acts as the nucleophile, and undergoes transmetalation to afford a palladium bis(aryl) complex. The latter complex can undergo reductive elimination to afford the cross-coupled product and regenerate the Pd(0) complex. In the absence of stronger electrophiles, aryl-bismuth species can directly undergo oxidative addition into the palladium catalyst, generating the corresponding oxidative addition complex and **Bi(I)** as a byproduct. If a nucleophile such as an organoboron compound is now present in the system, it can

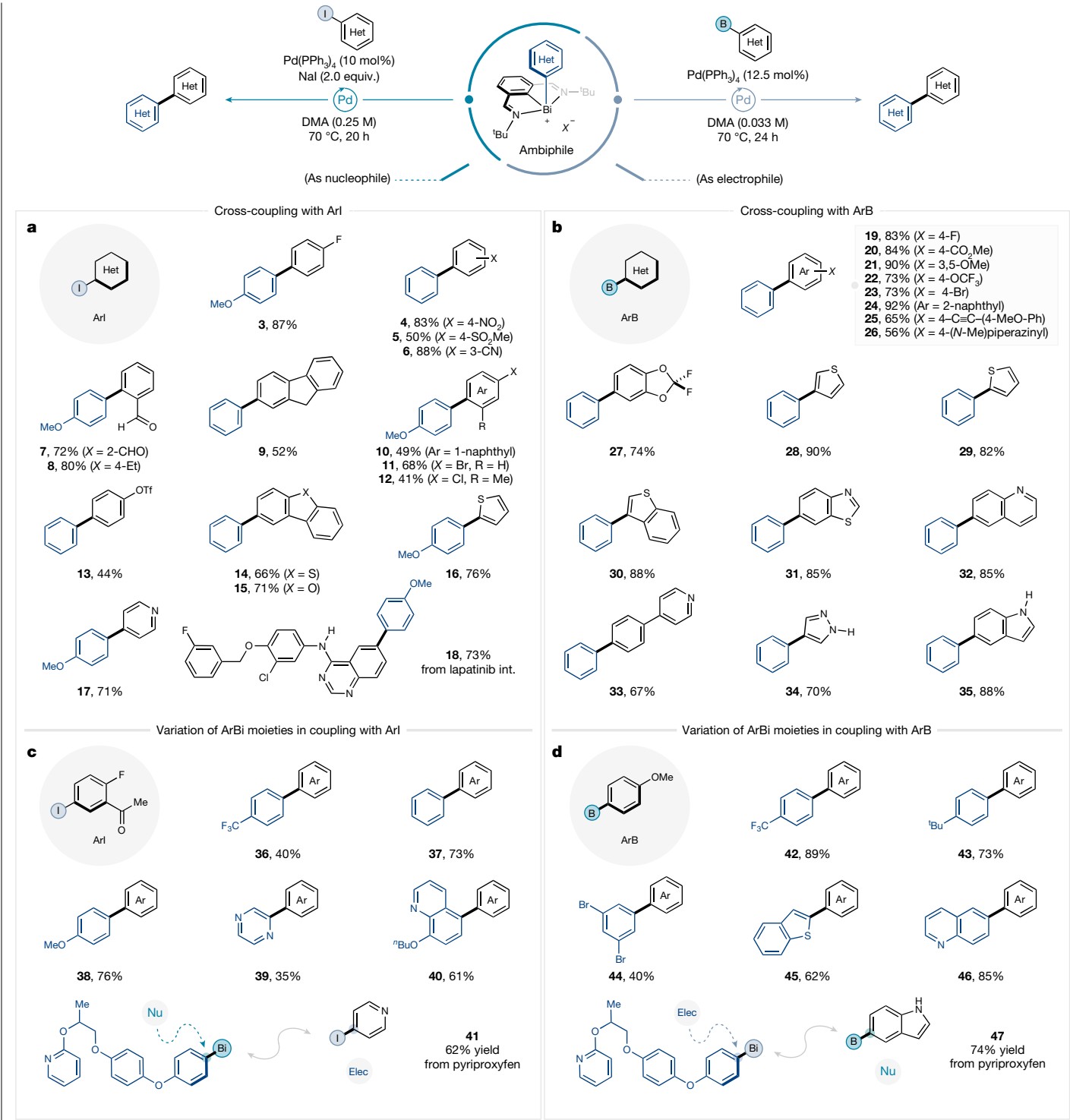

**Fig. 3 | Evaluation of aryl iodides and aryl boron compounds. a**, Ambiphilic cross-coupling of aryl-bismuth species (ArBi) as a nucleophile with aryl iodides (ArI); Ar = *p*-methoxyphenyl. **b**, Cross-coupling of ArBi as an electrophile with aryl boron reagents (ArB). **c**, Cross-coupling of structurally diverse aryl-bismuth compounds with aryl iodides. **d**, Cross-coupling of structurally diverse aryl-bismuth compounds with aryl boron reagents. Standard conditions for **a** and **c**: Pd(PPh₃)₄ (10 mol%), aryl-bismuth compounds (1.0 equiv.), aryl iodides (2.0 equiv.), NaI (2.0 equiv.), DMA (0.25 M), 70 °C, 20 h. Standard conditions for **b** and **d**: Pd(PPh₃)₄ (12.5 mol%), aryl-bismuth compounds (1.0 equiv.), aryl cyclic triol borates (2.0 equiv.), DMA (0.033 M), 70 °C, 24 h. Yields of isolated products.

then participate in the subsequent transmetalation–reductive elimination sequence, and thus furnish the cross-coupled product.

## Probing the generality of ambiphilic cross-coupling

After developing suitable catalytic conditions for both partners (Supplementary Information), the generality of the ambiphilic

cross-coupling was investigated with respect to electrophilic and nucleophilic cross-coupling partners (Fig. 3). For the ambiphilic coupling where the aryl-bismuth reagent behaves as the nucleophilic counterpart, aryl iodides were used as general electrophilic coupling partners (Fig. 3a). Importantly, this protocol can tolerate substituents *ortho*, *meta* and *para* to the C-I bond of the aryl iodide (**3–7**). Both electron-withdrawing (**4–7**) and electron-donating (**8**) groups

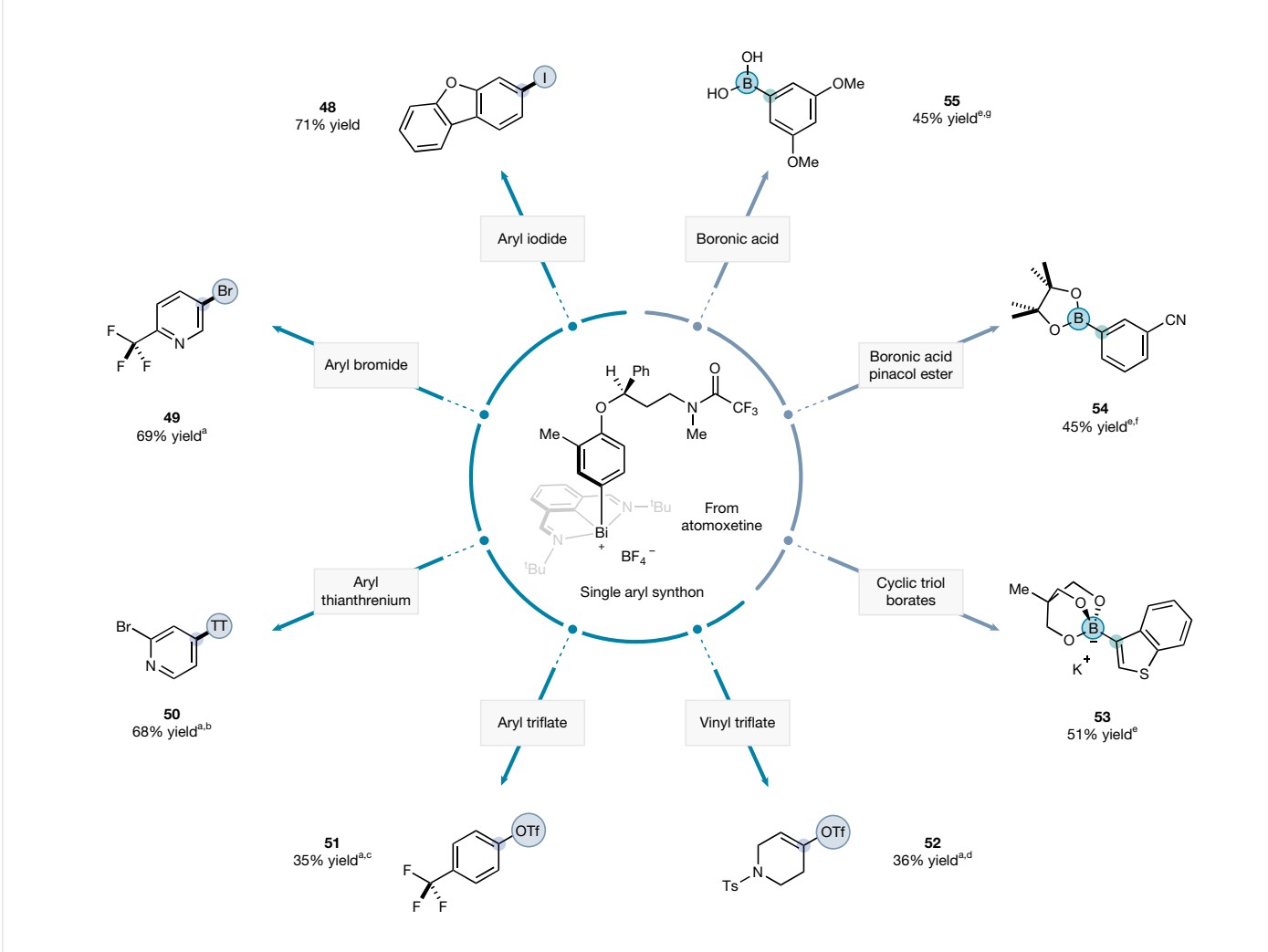

**Fig. 4 | Ambiphilic cross-coupling from a single aryl synthon: broad coupling partner compatibility.** [a]Conditions: Pd(TFA)$_2$ (12.5 mol%), PAd$_2$(*n*-Bu) (15.0 mol%), ArBr (2.0 equiv.), aryl-bismuth reagents (1.0 equiv.), NaI (10.0 equiv.), DMA (0.25 M), 70 °C, 20 h. [b]Aryl thianthrenium salt was used instead of aryl iodide. [c]Aryl triflate was used instead of aryl iodide. [d]Vinyl triflate was used instead of aryl iodide. [e]Conditions: Pd(PPh$_3$)$_4$ (12.5 mol%), aryl-bismuth reagents (1.0 equiv.), aryl cyclic triol borates (2.0 equiv.), DMA (0.033 M), 70 °C, 24 h. [f]Aryl boronic acid pinacol ester was used instead of aryl cyclic triol borates with K$_3$PO$_4$ (2.0 equiv.). [g]Aryl boronic acid was used instead of aryl cyclic triol borates with K$_3$PO$_4$ (2.0 equiv.). Yields of isolated products.

on the aryl iodide are well tolerated under the reaction conditions, affording the desired cross-coupled products in good to excellent yields, although aryl iodides bearing stronger electron-donating groups are not tolerated (for unsuccessful substrates, please see Supplementary Table 5). Various (pseudo)halides such as aryl bromide (**11**), aryl chloride (**12**) and aryl triflate (**13**) are compatible with the developed conditions and provide handles for further downstream functionalization. In addition, polyaromatics and pharmaceutically relevant heterocycles including fluorene (**9**), naphthalene (**10**), dibenzothiophene (**14**), dibenzofuran (**15**), thiophene (**16**) and pyridine (**17**) were also compatible with this protocol, affording the cross-coupled products in good yields. A densely functionalized aryl iodide bearing a secondary amine, chlorine and quinoxaline moieties (lapatinib intermediate) was smoothly converted to the corresponding cross-coupled product (**18**).

For the ambiphilic coupling where the aryl-bismuth compounds were used as electrophilic counterparts, structurally and electronically diverse aryl borates were investigated in the cross-coupling (Fig. 3b). The presence of either electron-withdrawing moieties such as methyl ester (**20**) and trifluoromethoxy (**22**) groups or electron-donating

methoxy (**21**) and piperazyl (**26**) groups was well tolerated under the reaction conditions. In addition, bromo- (**23**), naphthyl- (**24**) and alkynyl- (**25**) were also well tolerated functional groups. Furthermore, a wide variety of heterocyclic scaffolds prevalent in drug discovery efficiently underwent the cross-coupling, affording the corresponding heteroaryl compounds including benzodioxole (**27**), thiophene (**28** and **29**), benzothiophene (**30**), benzothiazole (**31**), quinoline (**32**), pyridine (**33**), pyrazole (**34**) and indole (**35**). Finally, heterocycles bearing acidic functionalities also afforded excellent yields of coupling products (**34** and **35**) without the use of an exogenous base, highlighting the robustness of the protocol. However, some protic functionalities including primary alcohols still remain challenging (Supplementary Table 5).

Finally, a range of structurally diverse ambiphilic aryl-bismuth reagents were evaluated under both sets of conditions (Fig. 3c,d). The cross-coupling with aryl-bismuth reagents bearing electron-withdrawing trifluoromethyl and electron-donating methoxy groups could transmetalate efficiently and afforded the desired products (**36**–**38**) (Fig. 3c). Heterocyclic aryl-bismuth reagents such as pyrazine (**39**) and quinoline (**40**) were also shown to be viable coupling partners. Similarly, it was demonstrated that an electronically diverse

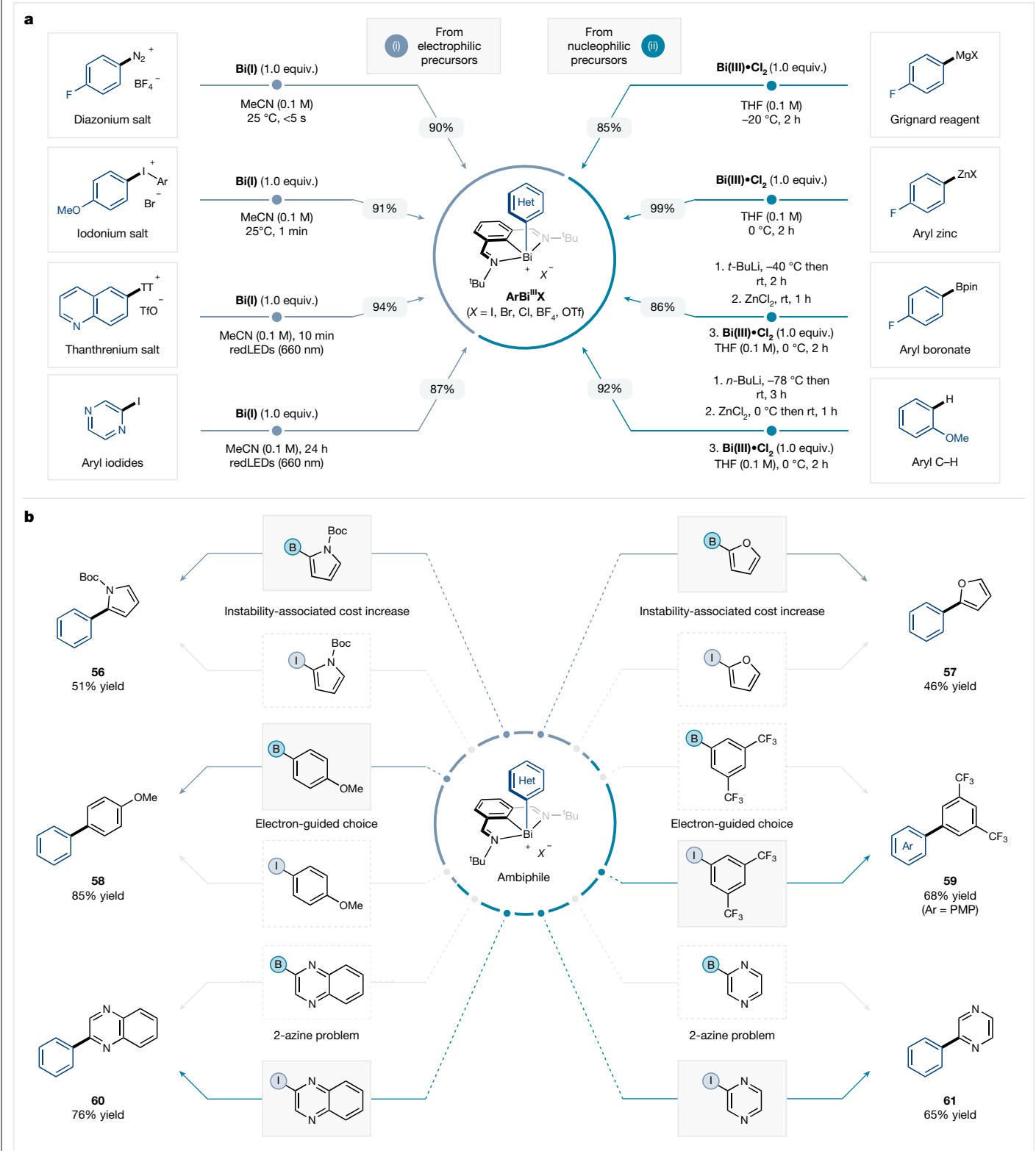

**Fig. 5 | Application of ambiphilic cross-coupling. a**, Access to ambiphilic aryl-bismuth reagents from nucleophilic and electrophilic aryl synthons. **b**, Strategic application of ambiphilic cross-coupling: case studies. rt, room temperature.

range of aryl-bismuth compounds could also be used as electrophilic coupling partners with aryl borates. Aryl bismuth species bearing electron-withdrawing or moderately electron-donating groups (**42** and **43**) and heteroaryl-bismuth species bearing benzothiophene (**45**) and quinoline (**46**) aryl groups could be cross-coupled in good yields. In addition, aryl-bismuth partners bearing functionalities susceptible

to oxidative addition could be employed in the reaction in moderate yield (**44**) (Fig. 3d). To showcase that aryl-bismuth reagents derived from densely functionalized and biologically active drug-like molecules were also competent ambiphiles, derivatives of pyriproxyfen (**41** and **47**) were shown to undergo cross-coupling smoothly with (hetero)aryl iodides and (hetero)aryl borates.

After validating the conceptual framework with the aryl iodides and aryl borates, we probed the limits of the ambiphilic cross-coupling strategy with a broader set of nucleophilic and electrophilic coupling partners (Fig. 4). Notably, the aryl bismuth derived from atomoxetine underwent cross-coupling with heteroaryl bromides (**49**), aryl thianthrenium salts (**50**), aryl triflate (**51**) and even vinyl triflate (**52**), and thus opened the door to a larger breadth of electrophiles, including C–H bond functionalization-derived partners (Fig. 4). Crucially, more accessible and commercially available aryl boronic acid pinacol ester (**54**) and aryl boronic acids (**55**) also served as competent nucleophilic counterparts, furnishing the corresponding cross-coupled products, although the presence of a base was required. Taken together, these results underscore the potential of the ambiphilic cross-coupling strategy: a single aryl-bismuth scaffold can be used to engage aryl iodides, thianthrenium salts, boronic acids, pinacol boronate esters, borate libraries and, in certain cases, aryl bromides, aryl triflates and vinyl triflates, without the need to prepare polarity-specific substrates.

## Strategic application of ambiphilic cross-coupling

As a result of the recently disclosed redox properties of organobismuth compounds and their application in catalysis[28–30], aryl-bismuth(III) compounds can be easily accessed from a plethora of distinct arylating precursors[31–35] (Fig. 5a). Notably, these protocols do not require the use of transition metals to forge the Bi–Ar bond. Aryl diazonium salts derived from corresponding aryl amines or iodonium salts arising from aryl iodides were readily converted to the corresponding ambiphilic aryl-bismuth reagents by a fast reaction with **Bi(I)** (Fig. 5a, top left). Both aryl thianthrenium salts and aryl iodides can also be converted to the ambiphilic aryl-bismuth species under red light irradiation (Fig. 5a, bottom left). Alternatively, aryl-bismuth compounds were efficiently accessed from Grignard reagents, arylzinc reagents and even aryl boronic acid derivatives by means of transmetalation with air-stable **Bi(III)** precursors (Fig. 5a, top and middle right). In addition, aryl-bismuth compounds can be easily synthesized by means of a C–H metalation procedure (Fig. 5a, bottom right)[36].

In addition, this strategy can also be harnessed to bypass inherent limitations in certain (hetero)aryl nucleophiles and electrophiles (Fig. 5b). For example, electron-rich 2-halo heteroaromatic reagents such as 2-iodopyrrole and 2-iodofuran are reported to be unstable[37,38], often being less accessible and more expensive commercially, whereas the corresponding boron analogues are widely available, bench stable and inexpensive (Supplementary Fig. 10). Indeed, the latter reagents could be employed in the cross-coupling to afford the desired products in good yields (**56** and **57**). In certain instances, sluggish oxidative addition is commonly faced when using electron-rich aryl halides, which thus makes these building blocks ineffective (Fig. 5b, middle)[39]. On the other hand, the corresponding organometallic reagent is more prone to fast transmetalation[40]. This is exemplified in the efficient cross-coupling reaction to afford product **58**, which was derived from an electron-rich boronic acid derivative instead of the corresponding aryl iodide. Conversely, the cross-coupling of the aryl-bismuth compound with an electron-deficient arene to afford **59** is more suited to an aryl iodide than to the boronic acid analogue. As such, this strategy allows the selection of a coupling partner based on the electronics of the arene, which thus enables efficient cross-coupling that is independent of the electronic nature of the arene. A major issue in cross-coupling is the use of α-substituted azines as nucleophiles[41]; for example, 2-pyrazyl boronic acid species are prone to rapid protodeborylation during couplings, which complicates their use as a nucleophile (Fig. 5b, bottom). As mentioned above, the current protocol allows the implementation of the stable heteroaryl iodide analogue, resulting in high yields of cross-coupling products (**60** and **61**). Collectively, these results provide a blueprint for ambiphilic cross-coupling as a versatile platform to construct $C(sp^2)$–$C(sp^2)$ bonds.

## Conclusion

In this work, we introduce the concept of ambiphilic cross-coupling, where a well-defined aryl-bismuth complex can serve as either an electrophilic or nucleophilic partner under transition-metal catalysis. Stoichiometric mechanistic studies revealed that transmetalation between aryl-bismuth compounds and Pd(II) aryl complexes can occur, leading to the formation of $C(sp^2)$–$C(sp^2)$ coupled products. Uniquely, these aryl-bismuth complexes can also react with Pd(0) by means of oxidative addition into the Bi–$C(sp^2)$ bond of the aryl-bismuth complex, leading to well-defined Pd(II) aryl complexes. Building on this mechanistic manifold, a catalytic ambiphilic cross-coupling protocol was demonstrated, wherein aryl-bismuth complexes can be coupled with a wide variety of electronically diverse (hetero)aryl iodides, bromides and boronic acid derivatives under mild conditions. Given the large (and growing) repertoire of methods that can be employed to access the aryl-bismuth complexes used in this study, this strategy circumvents issues encountered with a specific nucleophilic or electrophilic coupling partner. By overcoming traditional polarity constraints in reagent design, the ambiphilic cross-coupling strategy provides pronounced synthetic flexibility, enabling a single aryl synthon to engage with both nucleophilic and electrophilic coupling partners.

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

## Data availability

All data supporting the findings of this study are available within the main text and the Supplementary Information, or from the corresponding author upon request.

**Acknowledgements** We thank the analytical department at the MPI-Kohlenforschung for their support with compound characterization. We acknowledge M. Leutzsch for the characterization of bismuth complexes and A. Stamoulis for early contributions. We thank the members of the Cornella Group for proofreading. We are also grateful to A. Fürstner for generous support. The financial support for this work was provided by Max-Planck-Gesellschaft, Max-Planck-Institut für Kohlenforschung and by the Deutsche Forschungsgemeinschaft (DFG, German Research Foundation) under Germany's Excellence Strategy EXC 2033–390677874–RESOLV. This project also received funding from the European Union's Horizon 2020 research and innovation programme under agreement no. 850496 (ERC Starting Grant, J.C.).

**Author contributions** J.C. and B.R. conceptualized and designed the project. B.R. optimized the process, developed the approach, performed the experiments, analysed the experimental data, conducted the mechanistic studies and prepared the Supplementary Information. B.A.W. designed the synthesis of ambiphiles. All authors prepared the manuscript. J.C. directed the project.

**Funding** Open access funding provided by Max Planck Society.

**Competing interests** The authors declare no competing interests.

## Additional information

**Correspondence and requests for materials** should be addressed to Josep Cornella.
