## [Peer Review File · Nature]

Ambiphilic cross-coupling with aryl-bismuth reagents

Corresponding Author: Dr Josep Cornella

Version 2:

Reviewer comments:

Referee #1

(Remarks to the Author)

This manuscript by Cornella describes ambiphilic arylbismuth reagents that can undergo efficient Pd-catalyzed cross-coupling with either electrophiles (aryl iodides) or nucleophiles (aryl boronates). The authors show that this unusual ability arises from the reactivity of the arylbismuth species towards both oxidative addition of Pd(0) and transmetalation to Pd(II), depending on the conditions. The stoichiometric experiments proving this property are easy to understand and unambiguous. Importantly, no external oxidant or reductant is used, so the approach is conceptual distinct from “cross-electrophile,” “reductive,” or “oxidative” cross-coupling reactions that can couple nucleophiles/electrophiles with the opposite polarity. I agree that this method stands in contrast to the traditional paradigm that reagents must have one or the other intrinsic polarity in transition-metal catalyzed cross-couplings.

This paper has the simplicity, conceptual novelty, and “unobviousness” that characterize many compelling publications in Nature. If, as claimed, this is a new class of reagents that “liberate” cross-coupling from the constraints of intrinsic polarity, the work should be considered as groundbreaking. Some other advantages of the methods should be noted: the (pre)catalyst is simple and commercially available (tetrakis), and the conditions are mild. The demonstrated scope and functional-group tolerance are moderate (aryl iodides only, few protic FGs, few nitrogen heterocycles, etc.), but I feel it is mostly fine for a proof-of-concept paper (one exception noted below). Surely, the applicability will be expanded upon and improved later. I also felt that Fig. 5 is a very strong addition to the study.

I have two related reservations (maybe confusions) about this work's suitability for publication in its current form. Both might be addressable by the authors, either through revision or just a written response (in the case of my confusion). They are not necessarily disqualifying points but major enough to require consideration.

First, the ambiphilic paradigm could be described more precisely to allow readers to draw accurate comparisons with the current options. The idea is interesting and novel. But I am not convinced about “release from polarity-based thinking” and “one aryl synthon, double chemical space.” If instead of having nucleophiles and electrophiles, everything can just be an ambiphile, and the user can now couple any pair instead of one from each bin, that might deserve the “paradigm shift” and “liberation” label. After all, discarding the nucleophile and electrophile labels would create more chemical space from much fewer building blocks, and then I think the title of “ambiphilic cross-coupling” is appropriate. That is not quite what is accomplished here. Each coupling reaction still polar, not ambiphilic. The distinction is that one of the components (which is ambiphilic) can be used in either mode (the arylbismuth). So, in a discovery library, we still need many nucleophiles and electrophiles, plus some components (perhaps the complex or advanced intermediates) that can react with both.

We also have to consider that aryl nucleophiles and aryl electrophiles can generally be easily interconverted in a single step. Sometimes, as in reductive (cross-electrophile) coupling, the polarity can even be switched during the reaction. These arylbismuth compounds are themselves mostly synthesized from aryl nucleophiles or aryl electrophiles (there are some exceptions like thianthreniums), which takes a step or two also. So, what is the practical difference between converting some nucleophiles to electrophiles (and vice versa), or doing cross-electrophile couplings, compared to synthesizing some arylbismuth compounds? If we wanted to diversify some drug intermediate, what is the argument for synthesizing an arylbismuth compound (likely from an aryl halide or arylmetal reagent) versus, for example, just making half of it into arylboron and half into aryl halide? It is possible there are good answers, but the introduction has not spelled them out.

Basically, the authors need to illustrate exactly how they imagine these compounds being applied, and what is the advantage for users over the next best alternative. Only then can the potential impact be properly evaluated. Otherwise, I am afraid the idea “liberation ... from the polarity constraints that dominated the field” is not quite backed up by the results on their own.

Second, when comparing the ambiphilic “paradigm” to regular cross-coupling, it is critical to consider the scope of electrophiles and nucleophiles that can be paired with these ambiphiles. For example, it is not beneficial to have an ambiphilic reagent available that could only couple with 10% of the electrophiles/nucleophiles out there (especially if the ambiphile had to be synthesized from an electrophile or nucleophile to begin with). That is not really double the chemical space. I acknowledge that it is unreasonable to expect a proof-of-concept method to rival well-optimized catalytic systems in scope, so some limitations are to be expected.

The vast majority of nucleophile-mode coupling examples are with aryl iodide electrophiles, and this is a fairly severe limitation given the accessibility of iodides vs bromides/chlorides (about an order of magnitude). In addition, it is unclear if this limitation is just due to lack of optimization or something more fundamental – if the ambiphiles only undergo transmetalation with species such as $(\text{Ph}_3\text{P})_n\text{Pd}(\text{Ar})\text{X}$ but not those bearing more electron-rich phosphines, then it seems unlikely it can ever work with ArCl and many ArBr . Furthermore, the vast majority of the electrophile-mode coupling examples are with triolborates, not common boronic acids or pinacol esters. If there is now an addition (albeit easy) step required to make the triolborate, that undermines the generality and utility for most real-world users.

Minor comments:

- There are no examples with substitutions ortho to Bi. Is this a limitation of the method?
- The procedure for path G seems incorrect.
- ^{13}C NMR should be reported to 1 decimal place in most cases.
- Especially because these reactions are generally performed on very small scales, it is important that all compounds meet purity standards. Some solvent and grease, while not ideal, is fine, but compounds 18, 46, 47, 50, 51, 52 have enough “other stuff” in the NMR that I wonder about the accuracy of the gravimetric yield.

Referee #2

(Remarks to the Author)

The paper of Cornella and co-workers is very compelling for its conceptual simplicity: a single reagent that can participate in cross-coupling as either the electrophile or the nucleophile, depending on what it is partnered with. Arguably, aryl halides can be used to achieve the same net divergence by choice of appropriate reaction conditions (coupling with a nucleophile under standard cross-coupling conditions, or with an electrophile under reductive cross-coupling conditions). The present method is, however, clearly differentiated by the apparent ability of a single Bi fragment to act as both the electrofuge and the nucleofuge under such similar reaction conditions – this is proposed to result from the ambiphilic nature of the metal and the relative rates of the associated canonical steps, rather than due to introduction of an exogenous oxidant or reductant (cf XEC). As such, I am confident in the novelty of the concept, and that it is a significant advance (especially given how well explored the field of cross-coupling is).

This is definitely a concept driven paper (ie, it is not yet a practical competitor for other cross-couplings, including those that using Ar-Bi nucleophiles: Pd loadings are too high, yields are too low, scope is too narrow, and Bi reagents are not commercial). I do not see this as an issue at all, but in its current form the paper leaves too many questions unanswered: it needs to offer more insight, rather than just the observation and its exemplification. For example:

- Homocoupling is clearly a major competing process, eg, “...alongside 26% yield of homocoupling byproduct 2” (page 3). 2 is from homocoupling of ArI, rather than Ph-Bi, and is a major side-product in all the examples for which it is quantified. Its origin should therefore be explained/proposed in the manuscript, or the reader should be directed to a specific discussion in the SI. For example, does the Bi(I) co-product insert into ArI (presumably not without red light) to give Ar-Bi , which then couples with ArI to give 2? Or is Bi(I) acting as a reductant (as per XEC)?
- Related, does homocoupling of Ar-Bi occur? It would be very easy to explain (ie, OA into the Bi-C bond, followed by transmetalation of the Bi-C bond), and it would be quite surprising if this side-product was not formed. Given that reaction optimization was performed using ^{19}F NMR, Ph-Bi homocoupling could be overlooked in the majority of screening, but would be easy to observe with ^1H NMR or GC. The authors should state in the manuscript whether this side-product is formed at all, and adjust the footnote to Fig S9 if appropriate.
- Similarly, does protodeboronation/-debismuthation/-dehalogenation occur? If so, please state it in the manuscript. If protodebismuthation is not observed it could be an additional benefit vs boronic acids in standard Suzuki couplings.
- In the scope sections, please state whether homocoupling, protonation, or other side reactions (and of which fragment) were observed in a significant quantity (eg, above 5%). Where homocoupling was significant, please specify the amount for each entry in the SI. This important information because it speaks to the practical utility of the method, and may also reveal some additional mechanistic understanding in subsequent studies.
- Please also note the correct terminology for the homocoupled species is ‘side-product’, not ‘byproduct’. See: <https://pubs.acs.org/doi/10.1021/op300317g>
- P3: What is the role of NaI in the $\text{Ar-Bi} + \text{ArI}$ couplings? It clearly effects yield, but also apparently the cross-/homo-coupling selectivity (eg, Table S3: ~3:1 with NaI, vs ~8:1 without NaI). Please discuss in the manuscript or the SI.

- P3: “the lower yields were indicative of side reactivity of the Pd species” The mass balance of the fluorophenyl component doesn’t add up in most entries; what is the mass balance of the fluorophenyl, and of the remaining Bi reagent? This should be described in the manuscript or the SI.
- Similarly, P3: “PhBiIII₂Br was directly combined with Pd(PPh₃)₄, which furnished the Pd(II) oxidative addition complex along with the N,C,N-pincer bismuthinidene [Bi(I)] (Fig. 2B, i).” What is the mass balance of the ‘Ph’ (for example, do you see transmetallation from Bi, then RE to form Ph-Ph? Or protodebismuthation?)? This should be described in the manuscript or the SI.
- In Table S4, what is the role of light irradiation, if any? Is the apparent uplift in yield just a heating effect? If so, it may be worth noting this in the footnote of the table.
- Footnote to Table S5: “In both cases, a characteristic bismuth–ligand coupling byproduct—arising from C–C coupling between the 1,3-di-imine aryl ligand and the coupling partner, followed by hydrolysis to the corresponding 1,3-formyl derivative—was occasionally observed.” This side product should be mentioned in the manuscript, with the reader directed to more detail in the SI (which should cover: a proposal of how it is formed, the yield for each case it is observed). These side products (or representative examples) should be characterised, rather than just stated as fact with no supporting data.
- In terms of the mechanism for formation of this side-product (which should be discussed in the SI): do you think it is Pd-catalyzed, or a direct ligand coupling from a diaryl Bi(III) species?
- Discussion around these side products is clearly important as they are Bi-derived, and therefore unique to the ambiphilic coupling.
- Fig 3A: Presumably the choice of an electron-rich anisyl-Bi (or indeed a very electron-poor ArI in Fig 3C) is to maximise cross-selectivity? Please add examples to the manuscript to illustrate this by coupling Ph-Bi and 4-F₃CC₆H₄-Bi with 4-EtC₆H₄I and 4-FC₆H₄I.

Other points for consideration:

There are some points where the authors should be more balanced in the description of their work. At points the discussion is exaggerated or inaccurate, which distracts from the otherwise excellent science:

- P3: “product 1 is efficiently obtained in 79% yield...” please remove ‘efficiently’ – at 10 mol% Pd and with 2 equivalents of ArI it is not efficient (although it is high yielding), especially in comparison to other cross-couplings (including those based on Bi nucleophiles).
- P4: “and electron-donating (8) groups on the aryl iodide are well-tolerated (8) under the reaction conditions” This is inaccurate, verging on misleading. The ethyl substituent is a very modest electron donor, and stronger EDGs fail (as shown in Table S5). The discussion should say something like ‘modest electron-donating (8) groups on the aryl iodide are well-tolerated... although stronger electron donors are not tolerated (see Table S5)’.
- P5 “heterocycles bearing acidic functionalities also afforded excellent yields”. While this is not technically incorrect, it does imply that a broader range of acidic groups is tolerated, and greater clarity would be beneficial; the authors should direct the reader to the benzyl alcohol in Table S5, which does not work.
- P4: “this protocol can tolerate substituents ortho, meta, and para to the C–I bond (5–7).” Please also draw attention to failures here with a statement like ‘for unsuccessful substrates, please see Table S5’
- P5: “... piperazyl (26) groups posed no difficulties.” A yield of 56% using 12.5 mol% Pd and 2 equivalents of triolborate suggests that the piperazyl group does pose difficulties. Please consider your wording here – the hyperbole just distracts from what are otherwise impressive results.
- P5: “Good yields of cross-coupled products could be obtained with aryl-bismuth species bearing electron-withdrawing or -donating groups (42–44)” cpd 44 gives 40% yield; again, please consider your use of ‘good’ for the yield of this compound.
- P9 / Fig S6: the price comparisons between ‘equivalent’ aryl iodides/boronic acids are incorrect, and must be corrected:
 - o Fig 5B shows the NH pyrrole boronic acid but lists the price for the N-Boc analog. This is clearly not the same fragment as is used for the iodide, which is priced for the NH pyrrole. So either the figure must be adjusted to show the N-Boc boronic acid (and the direct comparison breaks down), or the price should be adjusted (The NH pyrrole boronic acid is available at BLD, 1 g / 786€, so the price differential is actually 1.3x more expensive for ArI, not 740x). Note that changing the figure to show the NH pyrrole boronic acid would be inappropriate because this substrate was unsuccessful, so this is not a viable ambiphilic coupling.
 - o choosing a 25g bottle for the Boc-pyrrole-boronic acid for the price comparison to avoid the ‘price inflation’ of a 1 g bottle, but the ‘inflated’ 1g bottle for the iodide is not appropriate. You should compare like masses to capture the same extent of ‘price inflation’.
 - o The unfavorable comparison of the iodofuran to the furyl boronic acid is really an issue that is specific to the ambiphilic coupling. If bromides could be coupled in a general way, the price comparison would be far less dramatic (1 g / 25 USD, oakwood).
- P9: “In certain instances, sluggish oxidative addition is commonly faced when using electron-rich aryl halides, thus making these building blocks ineffective (Fig. 4B, middle)³⁴. On the other hand, the corresponding organometallic reagent is more prone to fast transmetalation³⁴.” Some issues here:
 - o referencing an entire text book here is not appropriate; specific primary literature should be cited to support this point (ideally relating to the actual substrates you studied).
 - o Due to errors in figure and compound numbers it is not clear exactly which coupling partners you are discussing in this

section, but I presumably the 3,5-dimethoxy iodide and boronic acid. The authors will be aware that the meta-OMe is electron withdrawing, not donating, so this is not a good example to use as an electron rich aryl group; please either change the example, or provide primary literature that shows this aryl group to be specifically sluggish in OA, and fast in TM. An alternative, and more compelling argument to make with this particular fragment is that the iodide cannot be easily accessed by electrophilic iodination or lithiation/iodination, whereas the boronic acid is presumably easily accessible by C-H borylation.

• P9: “a single aryl-bismuth scaffold can explore aryl iodides, bromides... libraries” First, this anthropomorphises the Bi reagent – “a single aryl-bismuth scaffold can be coupled with / used to explore....” is more appropriate wording. Second, the scope with respect to bromides is very limited, so ‘exploration’ of this space is not realistic. From the presented scope, a single Bi reagent could be coupled with a subset of electron-poor ArBr only. Please consider rewording to make the phrasing more representative of the reality.

Other points for consideration:

- P2: “This conceptually divergent framework in cross-coupling has a profound effect in the exploration of chemical space: the preparation of one aryl synthon allows the indistinctive scrutiny of both aryl nucleophiles and electrophiles at the same time (Fig. 1B).” This sentence is not clear, and does not convey what I suspect the authors actually mean – please consider revising. “conceptually divergent framework”: framework is probably not the right word here (strategy?), and it is conceptually novel, rather than conceptually divergent. “indistinctive scrutiny of both aryl nucleophiles and electrophiles at the same time” I assume that the authors mean that the Bi reagent does not distinguish between / is not limited to a single class of coupling partners, and therefore that both ArI and ArB(OH)₂ can be used. But presumably both ArI and ArB(OH)₂ cannot be used simultaneously. (this may just be my misinterpretation, but I feel that this will also confuse other readers, so is worth reconsidering to maximise the impact of the work).
- Fig 1B: It would be helpful to draw the actual products of each coupling for clarity to the reader. And the meaning of “merged nucleophile & Electrophile chemical space” is not clear, please consider revising
- P3: more references are required for precedented Bi cross-couplings (seminal reports)
- Fig 3A: please replace “Ar” with the full structure in each case. There is space, and it will be far easier to interpret.
- P3: “aryl-bismuths”. ‘Bismuths’ is not a word; ‘aryl-bismuth reagents’?
- P5: Fig 5A is called out before fig 4, so these need to be renumbered and re-ordered
- P7: “Importantly, these protocols do not require the use of transition metals to forge the Bi–Ar bond.” The intention of this sentence is confusing, because (to me) I was not expecting TMs to be used, and few methods for making the alternative coupling partners use TMs either (Miyaura borylation and C-H borylation are the only common methods that use TMs; electrophilic halogenation, Gignard formation, C-H lithiation do not...). Also, the use of a TM would seem preferable to the use of diazonium salts or organolithium reagents for Ar-Bi synthesis, so unless more context can be added please consider editing this sentence. ...
- Fig 5A: “...increasing chemical space exploration” I’m not sure you can claim this, but you could claim that you can increase the number of classes of coupling partners that can be engaged with a single reagent. ‘increasing chemical space exploration’ feels inappropriate because the scope in ArBr is limited (so very few of the ‘>200,000 ArBr’ can actually be used), and presumably many of the ArBr that can be used will also be represented in the ArI pool (etc), so how much is the chemical space of viable coupling partners actually increased. Please consider a clearer way of wording this point.
- Fig 4 needs ‘A’ and ‘B’ in the panels. These are discussed in the main text, but are not present in the Fig.
- P9: “nucleophiles and electrophiles (Fig. 4B).” this should be Fig. 5B.
- In the last R&D paragraph it looks like all the compound numbers are incorrect (possibly confusion due to the incorrect figure numbers though), so I cannot comment on this discussion until referencing is corrected.
- P9: “one aryl synthon is able to scrutinize both nucleophile and electrophile chemical space at the same time”

Supplementary Information

- Multiplicity should be added for 19F NMR (unless acquired as 19F{1H}, in which case state it for each entry), and integrals are needed where there is more than 1 F environment per molecule.
- Each known compound should be compared to literature data and a ‘matches literature data’ statement added (with a reference). This is standard.
- Technically, 13C NMR chemical shifts should be reported to only 1 decimal point due to the inherent resolution of the technique (unless specific processing steps were taken).
- Otherwise, the data are complete and the spectra beautifully clean.

Version 3:

Reviewer comments:

Referee #1

(Remarks to the Author)

This revised version of the manuscript by Cornella and co-workers includes substantial additional experimentation to address the points raised by the two reviewers. I do not have any further technical comments, and the work is fine to be published now, after which I am sure Cornella and maybe others will further study and develop the system.

With respect to the suitability of the paper for Nature, it seems the topic of discussion is whether this result (that ArBi compounds can participate both oxidative aryl transfer to Pd or redox-neutral transmetallation) is (A) a finding likely to have impact on cross-coupling practically or academically, or (B) a surprising and non-intuitive organometallic fact but without much implication on cross-coupling? In the first review, my feeling had been somewhere in the middle of the spectrum, but

maybe closer to B. The result challenges the traditional electrophile/nucleophile paradigm in the same way that reductive cross-electrophile couplings did, and that is special. But it is unlikely to have comparable practical or academic impact to the cross-electrophile couplings because of how limited the results are in their current form. In that sense, this is closer to an organometallics paper than a cross-coupling paper.

Since then, I have read the other reviewer's comments and the authors' responses. Much of their analysis has merit. I am guessing many readers will agree with them, and others like me will be slightly skeptical about the impact on cross-coupling. After much reflection, I have decided that the former camp is likely to be bigger, and accordingly, I do not see any reason to hold up publication. The only way to know for sure is to come back to the paper in 5 years and see what effect it will have had. I am sure it is a landmark paper in Bi chemistry, but do not anticipate it will have similar impact in cross-coupling. It goes without saying that I would be happy to be wrong.

Referee #2

(Remarks to the Author)

The authors have addressed comprehensively each of the points raised by the referees. In particular, I feel that the extra detail around side-product formation and limitations is especially valuable: as the authors will appreciate, a new method can typically only fail once in a lab before it is disregarded, so tempering expectations is important. The added detail highlights the significant practical limitations of the method (including the significant effect of electronics on cross vs homo selectivity, the potential for Bi-pincer decomposition, phosphine arylation, the lack of tolerance for ortho substitution or acidic protons, and the poor-modest performance of aryl bromides and boronic acids), but not the conceptual novelty.

The detail into the side reactions is very useful for the expert, but may be confusing to a more general user. therefore I think that a very useful addition to the SI would be a 'user guide' to help other scientists who may wish to try the chemistry, summarising the limitations and spelling out in simple terms which classes of coupling partners are best suited (eg, coupling e-poor Ar-I and e-rich Ar-Bi gives high yield and minimal homocoupling; e-poor Ar-Bi are prone to homocoupling; Ar-Br may benefit from Ad2PBU as ligand, and give more/less homocoupling (?); ortho substitution is generally not tolerated on either partner; etc).

One point that needs further clarification:

The footnote to Figure S5 states "although the ligand-coupling product 4 was observed by GC/MS under Pd-free conditions (see Figure S15)". Figure S15 does not show the formation of product 4, or illustrate Pd-free conditions - please either revise the statement or figure accordingly.

Reviewer 1 (*referee comments in italics*; authors' replies in plain font)

This manuscript by Cornella describes ambiphilic arylbismuth reagents that can undergo efficient Pd-catalyzed cross-coupling with either electrophiles (aryl iodides) or nucleophiles (aryl boronates). The authors show that this unusual ability arises from the reactivity of the arylbismuth species towards both oxidative addition of Pd(0) and transmetalation to Pd(II), depending on the conditions. The stoichiometric experiments proving this property are easy to understand and unambiguous. Importantly, no external oxidant or reductant is used, so the approach is conceptually distinct from “cross-electrophile,” “reductive,” or “oxidative” cross-coupling reactions that can couple nucleophiles/electrophiles with the opposite polarity. I agree that this method stands in contrast to the traditional paradigm that reagents must have one or the other intrinsic polarity in transition-metal catalyzed cross-couplings. This paper has the simplicity, conceptual novelty, and “unobviousness” that characterize many compelling publications in Nature. If, as claimed, this is a new class of reagents that “liberate” cross-coupling from the constraints of intrinsic polarity, the work should be considered as groundbreaking. Some other advantages of the methods should be noted: the (pre)catalyst is simple and commercially available (tetrakis), and the conditions are mild. The demonstrated scope and functional-group tolerance are moderate (aryl iodides only, few protic FGs, few nitrogen heterocycles, etc.), but I feel it is mostly fine for a proof-of-concept paper (one exception noted below). Surely, the applicability will be expanded upon and improved later. I also felt that Fig. 5 is a very strong addition to the study. I have two related reservations (maybe confusions) about this work’s suitability for publication in its current form. Both might be addressable by the authors, either through revision or just a written response (in the case of my confusion). They are not necessarily disqualifying points but major enough to require consideration.

Response: We sincerely thank the reviewer for their positive and thoughtful evaluation of our manuscript.

Comment 1: *First, the ambiphilic paradigm could be described more precisely to allow readers to draw accurate comparisons with the current options. The idea is interesting and novel. But I am not convinced about “release from polarity-based thinking” and “one aryl synthon, double chemical space.” If instead of having nucleophiles and electrophiles, everything can just be an ambiphile, and the user can now couple any pair instead of one from each bin, that might deserve the “paradigm shift” and “liberation” label. After all, discarding the nucleophile and electrophile labels would create more chemical space from much fewer building blocks, and then I think the title of “ambiphilic cross-coupling” is appropriate. That is not quite what is accomplished here. Each coupling reaction still polar, not ambiphilic. The distinction is that one of the components (which is ambiphilic) can be used in either mode (the arylbismuth). So, in a discovery library, we still need many nucleophiles and electrophiles, plus some components (perhaps the complex or advanced intermediates) that can react with both.*

Response: We thank the reviewer for this important and insightful comment and we would like to provide clarification on this point. The reagents employed in traditional cross-coupling reactions are defined by the distinct canonical mechanistic steps they undergo, namely transmetalation or oxidative addition, which we categorize as nucleophiles and electrophiles, respectively. This behavior correlates with the bond polarity of the coupling partners, which is an intrinsic property of the molecule and can't be readily modified. In contrast, the aryl-bismuth reagents developed in this study exhibit ambiphilicity, a feature not observed in commonly employed aryl synthons (such as aryl boron, aryl silicon, aryl halides, or aryl thianthrenium salts). Specifically, these aryl-bismuth species are capable of engaging in both canonical mechanistic steps (oxidative addition and transmetalation). When employed in a cross-coupling reaction these aryl-bismuth species chemoselectively undergo only oxidative addition or transmetalation when paired with either a nucleophile or electrophile, respectively.

In other words, the ambiphiles reveal their “philicity” when facing another well-defined counterpart. This is a crucial point for this concept. As soon as you have an ambiphile in hand, you have the synthetic flexibility to select a coupling partner that is either a nucleophile or an electrophile. We completely agree that the coupling is still polar, of course, otherwise the cross selectivity would disappear. We also completely agree with the reviewer that selecting two ambiphiles from each bin would be highly advantageous; yet, if both partners are ambiphiles, a discriminating factor/agent between them would be required in order to obtain cross-selectivity, as there would be no mechanistic differentiation to selectively form the bis-aryl palladium complex required to form a C-C bond.

Accordingly, we have revised the discussion to clarify that our work does not propose a completely apolar cross-coupling framework, but rather introduces a previously unrecognized ambiphilic reagent class that can react with either nucleophilic or electrophilic partners. As the reviewer pointed out, we have also clarified the wording of “double chemical space”, since we agree this is probably not accurate. Our original intention derives from the previous discussion, where one ambiphile could target two distinct chemical spaces (electrophiles and nucleophiles). We have hence replaced the word “double chemical space” with “expanded pairing logic”.

We also agree with the reviewer that the use of terms such as “paradigm shift” may not be fully justified by the present dataset, due to the current limitations of the proof-of-concept coupling. Accordingly, we have removed this terminology throughout the manuscript.

***Comment 2:** We also have to consider that aryl nucleophiles and aryl electrophiles can generally be easily interconverted in a single step. Sometimes, as in reductive (cross-electrophile) coupling, the polarity can even be switched during the reaction. These arylbismuth compounds are themselves mostly synthesized from aryl nucleophiles or aryl electrophiles (there are some exceptions like thianthreniums), which takes a step or two also. So, what is the practical difference between converting some nucleophiles to electrophiles (and vice versa), or doing cross-electrophile couplings, compared to synthesizing some arylbismuth compounds? If we wanted to diversify some drug intermediate, what is the argument for synthesizing an arylbismuth compound (likely from an aryl halide or arylmetal reagent) versus, for example, just making half of it into arylboron and half into aryl halide? It is possible there are good answers, but the introduction has not spelled them out. Basically, the authors need to illustrate exactly how they imagine these compounds being applied, and what is the advantage for users over the next best alternative. Only then can the potential impact be properly evaluated. Otherwise, I am afraid the idea “liberation ... from the polarity constraints that dominated the field” is not quite backed up by the results on their own.*

Response: We thank the reviewer for their comments on the practical comparison between ambiphilic cross-coupling and established strategies. We agree that aryl nucleophiles and electrophiles can often be interconverted, and acknowledge that most aryl bismuth compounds described here are prepared from such starting points, with exceptions such as thianthrenium or Bpin derivatives (via direct C-H).

We would like to echo the fact that the application of these ambiphiles provided in this work represents a blueprint of reactivity and a proof-of-concept rather than aiming to replace the state-of-the-art cross-coupling methods at this point. However, in a prospective context, certain advantages can already be devised. As acknowledged by this reviewer in their initial assessment, Figure 5 delineates a few examples where an ambiphilic approach would be advantageous. Indeed, although many aryl halides are of course readily interconvertible, there are significant examples where the corresponding aromatic boron species is unstable (2-pyridyl boronic acids) and the parent halide would be much preferred. We also highlight cases where the converse also holds true. In these circumstances where interconversion is challenging, converting the aryl synthon into an ambiphile would present a sound, viable solution.

From a practicality standpoint, these ambiphilic aryl-bismuth species are isolable and often even bench-stable, accessible from a wide variety of synthetic precursors. A stable, isolable ambiphile provides the flexibility to scout out future steps from the same point in a synthetic sequence, without having to

explicitly plan for specifically an electrophile or a nucleophile as the complementary partner. In this manner, ambiphiles can be prepared, stored and revisited later on without having to interconvert to the adjusted philicity required at every step.

In order to more accurately describe the distinctive features offered by ambiphilic scaffolds as opposed to preparing complementary aryl boron/aryl halide sets, we have modified the text of the manuscript accordingly:

'Isolable, bench stable aryl-bismuth reagents provide a modular platform for coupling reactions: in diversification of complex or high value aryl fragments they prevent repeated re-functionalisation steps in the interconversion between nucleophiles and electrophiles, thus enabling expedient parallel synthesis across both electrophiles and nucleophiles from a single intermediate'.

The use of words like 'liberation' have been removed.

Comment 3: *Second, when comparing the ambiphilic "paradigm" to regular cross-coupling, it is critical to consider the scope of electrophiles and nucleophiles that can be paired with these ambiphiles. For example, it is not beneficial to have an ambiphilic reagent available that could only couple with 10% of the electrophiles/nucleophiles out there (especially if the ambiphile had to be synthesized from an electrophile or nucleophile to begin with). That is not really double the chemical space. I acknowledge that it is unreasonable to expect a proof-of-concept method to rival well-optimized catalytic systems in scope, so some limitations are to be expected.*

Response: We thank the reviewer for this comment. We are in agreement with the reviewer on this matter. As pointed out above, the term "double" was probably not accurate, as we meant that one can target a "dual" philicity chemical space. As the reviewer points out, this concept is at a proof-of-concept stage, and development of a brand-new cross coupling that directly competes with the established methods would be inherently difficult. However, we believe that the unique reactivity displayed by the aryl bismuth reagents in the ambiphilic cross-coupling has the potential to streamline access to chemical space, due to the breadth of compatible coupling partners. As mentioned, 'doubling' has been substituted by 'expanded pairing logic' in Fig 1B.

Comment 4: *The vast majority of nucleophile-mode coupling examples are with aryl iodide electrophiles, and this is a fairly severe limitation given the accessibility of iodides vs bromides/chlorides (about an order of magnitude). In addition, it is unclear if this limitation just due to lack of optimization or something more fundamental – if the ambiphiles only undergo transmetalation with species such as (Ph₃P)_nPd(Ar)X but not those bearing more electron-rich phosphines, then it seems unlikely it can ever work with ArCl and many ArBr. Furthermore, the vast majority of the electrophile-mode coupling examples are with triolborates, not common boronic acids or pinacol esters. If there is now an addition (albeit easy) step required to make the triolborate, that undermines the generality and utility for most real-world users.*

Response: We thank the reviewer for this constructive comment. As noted, the initial scope of the ambiphilic coupling primarily employed aryl iodides as electrophiles and triolborates as nucleophilic partners. With respect to electrophile scope, while reactivity with aryl chlorides remains limited at present, we found that (hetero)aryl bromides can participate effectively under modified conditions employing an electron-rich phosphine ligand (PAd₂ⁿBu) (Figure. 4). These results indicate that the observed preference for aryl iodides reflects the current state of optimization rather than an inherent limitation of the ambiphilic system. Based on the comment of the reviewer, we further explored the reactivity beyond aryl halides and discovered that electron-deficient aryl triflates as well as vinyl triflates can be accommodated, thereby broadening the electrophile scope. These newly obtained results have been integrated into Figure 4. Regarding the

nucleophilic coupling partners, although the initial examples employed triolborates, we have also identified conditions under which commercially available boronic acids and boronic acid pinacol esters undergo efficient coupling using K_3PO_4 as an additive (**54**, **55**). We believe that these results expand the practical scope of the method and eliminate the need for prior conversion to triolborates, thereby enhancing its generality and synthetic utility.

Fig. 4. Ambiphilic cross-coupling from a single aryl synthon: broad coupling partner compatibility. ^aConditions: $Pd(TFA)_2$ (12.5 mol%), $PAd_2(n-Bu)$ (15.0 mol%), $ArBr$ (2.0 equiv), aryl-bismuth reagents (1.0 equiv), NaI (10.0 equiv), DMA (0.25 M), 70 °C, 20 h. ^bAryl thianthrenium salt was used instead of aryl iodide. ^cAryl triflate was used instead of aryl iodide. ^dVinyl triflate was used instead of aryl iodide. ^eConditions: $Pd(PPh_3)_4$ (12.5 mol%), aryl-bismuth reagents (1.0 equiv), aryl cyclic triol borates (2.0 equiv), DMA (0.033 M), 70 °C, 24 h. ^fAryl boronic acid pinacol ester was used instead of aryl cyclic triol borates with K_3PO_4 (2.0 equiv). ^gAryl boronic acid was used instead of aryl cyclic triol borates with K_3PO_4 (2.0 equiv). Yields of isolated products are indicated.

Comment 5: • *There are no examples with substitutions ortho to Bi. Is this a limitation of the method?*

Response: We thank the reviewer for the comment. *Ortho*-substituted aryl-bismuth reagents could be prepared from either anisole or 2-bromoanisole (Path F and H), but their reactivity in the present coupling context had not been assessed. We therefore evaluated representative *ortho*-substituted substrates under the standard conditions, obtaining only modest yields (up to 21%). The data have been added to *Table S5* in the supporting information.

Comment 6: • *The procedure for path G seems incorrect.*

Response: We thank the reviewer for bringing this to our attention and this has been duly corrected.

Comment 7: • *^{13}C NMR should be reported to 1 decimal place in most cases.*

Response: We thank the reviewer for pointing out this oversight. We have carefully reviewed the Supporting Information in its entirety and revised the reported ^{13}C NMR chemical shifts to be presented consistently to one decimal place.

Comment 8: • *Especially because these reactions are generally performed on very small scales, it is important that all compounds meet purity standards. Some solvent and grease, while not ideal, is fine, but compounds 18, 46, 47, 50, 51, 52 have enough "other stuff" in the NMR that I wonder about the accuracy of the gravimetric yield.*

Response: We thank the reviewer for this important comment regarding compound purity and yield accuracy. We re-examined the NMR spectra of the compounds in question and carried out additional purification in all cases. Re-purification afforded cleaner materials in several instances; the isolated product amounts were re-weighed and the gravimetric yields recalculated accordingly. The revised values have been updated in the manuscript and Supporting Information. For compounds **18**, **51**, and **52**, complete separation from minor side products could not be achieved despite re-purification. The residual impurities are small (typically $<5\%$ by ^1H NMR) and are not expected to significantly affect the reported yields; nevertheless, to ensure clarity for the reader, these compounds are described in the Supporting Information as being isolated "with small amounts of inseparable impurities".

Reviewer 2 (*referee comments in italics; authors' replies in plain font*)

***Overall:** The paper of Cornella and co-workers is very compelling for its conceptual simplicity: a single reagent that can participate in cross-coupling as either the electrophile or the nucleophile, depending on what it is partnered with. Arguably, aryl halides can be used to achieve the same net divergence by choice of appropriate reaction conditions (coupling with a nucleophile under standard cross-coupling conditions, or with an electrophile under reductive cross-coupling conditions). The present method is, however, clearly differentiated by the apparent ability of a single Bi fragment to act as both the electrofuge and the nucleofuge under such similar reaction conditions – this is proposed to result from the ambiphilic nature of the metal and the relative rates of the associated canonical steps, rather than due to introduction of an exogeneous oxidant or reductant (cf XEC). As such, I am confident in the novelty of the concept, and that it is a significant advance (especially given how well explored the field of cross-coupling is). This is definitely a concept driven paper (ie, it is not yet a practical competitor for other cross-couplings, including those that using Ar-Bi nucleophiles: Pd loadings are too high, yields are too low, scope is too narrow, and Bi reagents are not commercial). I do not see this as an issue at all, but in its current form the paper leaves too many questions unanswered: it needs to offer more insight, rather than just the observation and its exemplification.*

Response: We thank the reviewer for the positive overall assessment and for recognizing the conceptual novelty of this work.

***Comment 1:** • Homocoupling is clearly a major competing process, eg, "...alongside 26% yield of homocoupling byproduct 2" (page 3). 2 is from homocoupling of ArI, rather than Ph-Bi, and is a major side-product in all the examples for which it is quantified. Its origin should therefore be explained/proposed in the manuscript, or the reader should be directed to a specific discussion in the SI. For example, does the Bi(I) co-product insert into ArI (presumably not without red light) to give Ar-Bi, which then couples with ArI to give 2? Or is Bi(I) acting as a reductant (as per XEC)?*

Response: We thank the reviewer for this constructive and helpful comment. Homocoupling was consistently observed as a competing process in cross-coupling reactions involving aryl iodides. Guided by the reviewer's suggestions, we investigated the origin of this side product by examining the reactivity of Bi(I) and Bi(III) species that could be formed as reaction side products. Control experiments showed that, in the absence of aryl-bismuth reagents, only trace homocoupling was observed under the standard reaction conditions. Similarly, aryl-Bi(III) species in the absence of aryl iodide resulted in negligible formation of the homocoupling product.

In contrast, addition of Bi(I) species as an external additive led to efficient homocoupling. These results suggest that Bi(I) can act as a reductant, promoting palladium-catalyzed reductive homocoupling of the aryl iodide, consistent with an XEC-type pathway. A detailed discussion of these control experiments is now provided in the Supporting Information (*Figure S3*) and are referenced in the main manuscript.

Figure S3. Control experiments on the origin of aryl iodide-derived homocoupling products

Note: The results of the control experiments suggest that Bi(I) can act as a reductant, thereby promoting palladium-catalyzed reductive homocoupling of the aryl iodide.

Comment 2: • Related, does homocoupling of Ar-Bi occur? It would be very easy to explain (ie, OA into the Bi-C bond, followed by transmetalation of the Bi-C bond), and it would be quite surprising if this side-product was not formed. Given that reaction optimization was performed using ¹⁹F NMR, Ph-Bi homocoupling could be overlooked in the majority of screening, but would be easy to observe with ¹H NMR or GC. The authors should state in the manuscript whether this side-product is formed at all, and adjust the footnote to Fig S9 if appropriate.

Response: We thank the reviewer for this helpful comment. To clarify whether Ar-Bi homocoupling occurs, we analyzed the reaction mixtures by GC/MS, as identification by ¹H NMR spectroscopy is difficult as a result complex signal overlap in the aromatic region. Under the standard coupling conditions; only the desired cross-coupling product, the aryl iodide-derived homocoupling product and residual starting materials were predominantly detected. No aryl-bismuth-derived homocoupling was observed. In contrast, reactions conducted with the aryl-bismuth reagent alone showed clear formation of aryl-bismuth homocoupling

products, along with ligand-derived coupling products. These homocoupling products were isolated and fully characterized by ^1H NMR, ^{13}C NMR, and HRMS. Taken together, these results indicate that aryl-bismuth homocoupling is possible but does not significantly compete under the standard coupling conditions employed in this study. The corresponding full characterization data have been added to the Supporting Information as *Figure S5*, the footnote to *Figure S15* (previous Figure S9) has been revised accordingly and a brief description has been included in the main manuscript.

Figure S5. Observation of aryl bismuth-derived coupling products

A. Standard reaction conditions

B. Reaction without aryl iodides

Note: These observations indicate that aryl-bismuth species can engage in ambiphilic coupling behavior, acting as both a nucleophilic aryl donor and an electrophilic coupling partner. Consistent with this interpretation, aryl bismuth-derived coupling products (3 and 4) are observed in the absence of aryl iodide, supporting the possibility of off-cycle reactivity (see Figure S15).

Comment 3: • Similarly, does protodeboronation/-debismuthation/-dehalogenation occur? If so, please state it in the manuscript. If protodebismuthation is not observed it could be an additional benefit vs boronic acids in standard Suzuki couplings.

Response: We thank the reviewer for this helpful comment. To address this point, we reinvestigated the ambiphilic cross-coupling reactions using ^{19}F NMR spectroscopy as well as GC/MS and LC/MS analyses. The dehalogenation product was not detected under the standard reaction conditions. Protodeboronation was observed only in trace amounts by GC/MS. In contrast, a minor amount of (proto)-debismuthation product (compound 5 in response to Comment 2) was observed in selected cases.

Comment 4: • In the scope sections, please state whether homocoupling, protonation, or other side reactions (and of which fragment) were observed in a significant quantity (eg, above 5%). Where homocoupling was significant, please specify the amount for each entry in the SI. This important information because it speaks to the practical utility of the method, and may also reveal some additional mechanistic understanding in subsequent studies.

Response: We thank the reviewer for this important comment. Overall, two classes of side products derived from the aryl-bismuth reagents were observed: (1) Ar-Ar homocoupling originating from the aryl-bismuth reagent (ArBi and ArBi), and (2) cross-coupling involving fragments derived from the *N,C,N*-pincer ligand backbone with the external aryl coupling partner.

In contrast, homocoupling originating from the other *coupling partner* (rather than from the aryl-bismuth reagent) depended strongly on partner identity. For aryl electrophiles (e.g., aryl iodides), electrophile-derived homocoupling products were sometimes observed in appreciable amounts (typically ~5–15% in selected cases, see Response to Comment 2). By comparison, under the standard conditions the homocoupling of the aryl boron reagents was negligible and generally below the detection limit.

To address the reviewer's request for quantitative disclosure, we have now added the measured amounts of homocoupling/side products to the Supporting Information for representative substrates where these processes were $\geq 5\%$. While a fully systematic quantification for every entry in the scope was not performed, these data capture the dominant side-product pathways and provide practical guidance on reaction utility.

Figure S6. Systematic analysis of side product profiles across representative substrates.

Note: In cross-couplings of aryl-bismuth reagents with aryl electrophiles, electrophile-derived homocoupling was occasionally observed (up to ~15%). In cases where the homocoupled side product exhibits similar polarity to the

desired product, chromatographic separation can be challenging.

Comment 5: • Please also note the correct terminology for the homocoupled species is 'side-product', not 'byproduct'. See: <https://pubs.acs.org/doi/10.1021/op300317g>

Response: This has been duly changed in the main text.

Comment 6: • P3: What is the role of NaI in the Ar-Bi + ArI couplings? It clearly effects yield, but also apparently the cross-/homo-coupling selectivity (eg, Table S3: ~3:1 with NaI, vs ~8:1 without NaI). Please discuss in the manuscript or the SI.

Response: We thank the reviewer for this comment. A range of additives (NaI, NMe₄I, Na₂CO₃, K₂CO₃, K₃PO₄) were examined (Table S3), and all led to improved efficiency relative to additive-free conditions, with NaI affording the highest yield of the cross-coupled product. Although no clear or consistent trend in cross-/homocoupling selectivity was observed across the additives, the overall enhancement in reactivity suggests that additives promote activation of the aryl-bismuth reagent, facilitating its participation in the transmetalation step, analogous to base-assisted activation of organoboron reagents in Suzuki couplings. Minor changes in selectivity are tentatively attributed to partial promotion of competing homocoupling pathways, potentially influenced by the extent of Bi(I) species formation under additive-assisted conditions, consistent with our observation in Comment 2 that Bi(I) can act as a reducing agent. These points are now discussed in Figure S4 of the Supporting Information.

Figure S4. Additive effect

Note: Although the precise influence of individual additives on cross-/homocoupling selectivity remains unclear, their overall effect is proposed to arise from activation of the aryl-bismuth reagent, analogous to base-assisted activation of organoboron species in Suzuki coupling.

Comment 7: • P3: “the lower yields were indicative of side reactivity of the Pd species” The mass balance of the fluorophenyl component doesn’t add up in most entries; what is the mass balance of the fluorophenyl, and of the remaining Bi reagent? This should be described in the manuscript or the SI. • Similarly, P3: “PhBi(III)Br was directly combined with Pd(PPh₃)₄, which furnished the Pd(II) oxidative addition complex along with the N,C,N-pincer bismuthinidene [Bi(I)] (Fig. 2B, i.)” What is the mass balance of the ‘Ph’ (for example, do you see transmetallation from Bi, then RE to form Ph-Ph? Or protodebismuthation?)? This should be described in the manuscript or the SI.

Response: We thank the reviewer for this helpful comment. To address the mass balance issues raised for the reactions described on P3, we performed additional stoichiometric experiments and analyses. To probe the mass balance of the fluorophenyl component, we re-conducted stoichiometric reactions between the aryl-bismuth reagent and 1.0 equiv of the oxidative addition palladium complex, followed by GC/MS analysis. In these experiments, the Ph–Ph homocoupling product derived from aryl–bismuth, as well as the desired cross-coupling product, were consistently observed. Notably, a C–P cross-coupling product involving the PPh₃ ligand was also detected, which was confirmed by ¹⁹F NMR spectroscopic analysis showing a characteristic resonance at ca. –99 ppm across these reactions. Comparison with reported data suggests phosphonium salt, presumably formed via C–P cross-coupling between a fluorophenyl fragment and the phosphine ligand (Lee, Y. H., Morandi, B. *Nat. Chem.* **10**, 1016–1022 (2018)). On the basis of these observations, we propose that a significant portion of the fluorophenyl mass balance is accounted for by this side pathway.

A similar outcome was observed when examining the mass balance of the phenyl group in the stoichiometric reaction between PhBi(III)Br and Pd(PPh₃)₄. To address this point, we carried out additional NMR analyses. These experiments revealed the formation of a substantial amount of a phosphonium salt, which was fully characterized by NMR spectroscopy. Analogous to the fluorophenyl case, this product is attributed to a C–P cross-coupling side pathway and accounts for a major fraction of the phenyl mass balance. All newly acquired data supporting these conclusions have been added to the Supporting Information.

Figure S13. ^{19}F NMR spectra of reaction mixtures for entries 2 and 7.

Figure S14. Structures of products observed by GC/MS and reported phosphonium salts for comparison.⁹⁰

Note: GC/MS analysis revealed a common species present in both the cross-coupled product, the Ph-Ph homocoupling product derived from the aryl-bismuth reagents, and an unexpected C-P coupling product.⁹¹ Consistently, ^{19}F NMR analysis showed a signal near -99 ppm. Comparison with reported ^{19}F NMR data (377 MHz, CDCl_3 , δ -98.50 (d, $J = 1.3$ Hz) for $\text{Ph}_3\text{P}(4\text{-F-Ph})\text{Cl}$, reference 90) supports tentative assignment of this species as a phosphonium salt, likely formed via C-P cross-coupling between an aryl fragment and PPh_3 derived from $\text{Pd}(\text{PPh}_3)_4$. Related observations were also made in subsequent experiments examining oxidative addition between aryl-bismuth reagents and $\text{Pd}(\text{PPh}_3)_4$ (see Figure 16).

Figure S17. Oxidative addition complexes from the reaction of ArBi(Br)-1 with $\text{Pd}(\text{PPh}_3)_4$

Note: See Figure S24 to S30 for details on the characterization of phosphonium salts.

Figure S24. ¹H NMR spectrum of the phosphonium salt side product formed via C–P coupling

Figure S25. COSY spectrum of the phosphonium salt side product formed via C–P coupling

Figure S26. ^{13}C NMR spectrum of the phosphonium salt side product formed via C-P coupling

Figure S27. HMBC (^1H - ^{13}C) spectrum of the phosphonium salt side product formed via C-P coupling

Figure S28. HSQC (^1H - ^{13}C) spectrum of the phosphonium salt side product formed via C-P coupling

Figure S29. ^{31}P NMR spectrum of the phosphonium salt side product formed via C-P coupling

Figure S30. HMBC (^1H - ^{31}P) spectrum of the phosphonium salt side product formed via C-P coupling

Comment 9: • In Table S4, what is the role of light irradiation, if any? Is the apparent uplift in yield just a heating effect? If so, it may be worth noting this in the footnote of the table.

Response: We thank the reviewer for this comment. Based on our control experiments, the effect of light irradiation appears to be largely attributable to a thermal (heating) effect rather than a specific photochemical contribution. No distinct reactivity trends consistent with a light-induced pathway were observed beyond the increase in temperature. Accordingly, this interpretation has now been noted in the footnote of Table S4.

“Note: Under typical reaction conditions, the predominant species observed after the reaction is the cross-coupling product **I**, while in contrast to the previous case (cross-coupling with aryl iodides), no homocoupling product is observed. Furthermore, light irradiation, which was employed during early optimization studies, was found to exert only a thermal (heating) effect rather than promoting a distinct photochemical pathway.”

Comment 10: • Footnote to Table S5: “In both cases, a characteristic bismuth–ligand coupling byproduct—arising from C–C coupling between the 1,3-di-imine aryl ligand and the coupling partner, followed by hydrolysis to the corresponding 1,3-formyl derivative—was occasionally observed.” This side product should be mentioned in the manuscript, with the reader directed to more detail in the SI (which should cover: a proposal of how it is formed, the yield for each case it is observed). These side products (or representative examples) should be characterised, rather than just stated as fact with no supporting data. • In terms of the mechanism for formation of this side-product (which should be discussed in the SI): do you think it is Pd-catalyzed, or a direct ligand coupling from a diaryl Bi(III) species? • Discussion around these side products is clearly important as they are Bi-derived, and therefore unique to the ambiphilic coupling.

Response: We thank the reviewer for this important comment. The bismuth-ligand coupling side product was occasionally observed, however its formation was typically negligible under standard catalytic conditions. Notably, this side product was formed more prominently in the absence of an external coupling partner (e.g., aryl iodide or aryl boronate), as discussed in *our response to Comment 2 (Figure S5)*.

However, as expected from the poor ligand-ligand coupling ability of Bi(III), no formation of the cross-coupled product **2** was observed in the absence of the Pd catalyst (Figure S5A). Instead, in the absence of both the aryl iodide coupling partner and palladium, the aryl-ligand coupling side product **4** was detected by GC, albeit only in negligible amounts (<5% yield), corresponding to 18% conversion of the starting aryl-bismuth complex, with no formation of the aryl-aryl homocoupling product **3** (Figure S5B). These results indicate that this side reaction proceeds far more efficiently under palladium catalysis. Based on these observations, we propose that the side product primarily arises from Pd-mediated aryl transfer involving the aryl–bismuth reagent, consistent with its ambiphilic character, which enables both nucleophilic and electrophilic reactivity. These findings support the mechanism proposed in Figure S15 (previously Figure S9).

In addition, representative bismuth–ligand coupling products were isolated and fully characterized by ¹H and ¹³C NMR spectroscopy as well as HRMS analysis, and the corresponding characterization data have been added to the Supporting Information. Furthermore, this side reactivity is now described in the main manuscript as follows: “In addition, a characteristic bismuth–ligand coupling side product was occasionally observed, presumably arising from C–C coupling between the aryl group bound to bismuth and the aryl backbone of the N,C,N pincer ligand, further supporting the ambiphilic reactivity of the aryl–bismuth complex (see Figure S5 for details).”

Figure S5. Observation of aryl bismuth-derived coupling products

A. Standard reaction conditions

B. Reaction without aryl iodides

Note: These observations indicate that aryl-bismuth species can engage in ambiphilic coupling behavior, acting as both a nucleophilic aryl donor and an electrophilic coupling partner. Consistent with this interpretation, aryl bismuth-derived coupling products (3 and 4) are observed in the absence of aryl iodide, supporting the possibility of off-cycle reactivity, although the ligand-coupling product 4 was observed by GC/MS under Pd-free conditions (see Figure S15).

Figure S15. Plausible mechanism in stoichiometric experiment of OAC-1 with aryl-bismuth.

Note: In stoichiometric reactions between aryl-bismuth and the oxidative addition complex OAC-1, the reactivity was markedly diminished compared to the catalytic process. The addition of 2.0 equiv. of aryl iodide, however, restored the reactivity, leading to product formation in yields comparable to the catalytic system (80%). These findings suggest competitive side reactions. Specifically, after transmetalation of aryl-bismuth 1 to OAC-1, the cross-coupled product is generated, with concomitant formation of Pd(PPh₃)₂. If aryl-bismuth also undergoes oxidative addition with Pd(PPh₃)₂ acting as the electrophile, the overall efficiency may be reduced. Based on this hypothesis, aryl-bismuth

could act as an electrophile and undergo oxidative addition with palladium (see Figure S5). We subsequently carried out additional experiments to examine this possibility.

Comment 11: • Fig 3A: Presumably the choice of an electron-rich anisyl-Bi (or indeed a very electron-poor ArI in Fig 3C) is to maximise cross-selectivity? Please add examples to the manuscript to illustrate this by coupling Ph-Bi and 4-F3CC6H4-Bi with 4-EtC6H4I and 4-FC6H4I.

Response: We thank the reviewer for this helpful suggestion. As anticipated, electron-rich aryl-bismuth reagents generally behave as more effective nucleophilic partners in these cross-coupling reactions, presumably by facilitating the transmetalation step in a manner analogous to Suzuki–Miyaura coupling. To illustrate this effect, we examined the coupling of 4-ethylphenyl iodide and 4-fluorophenyl iodide with a series of aryl-bismuth reagents bearing different electronic properties (4-OMe, 4-H, and 4-CF₃). In the case of 4-fluorophenyl iodide, the desired cross-coupled product was obtained in 87%, 68%, and 45% yield when using 4-OMe-, 4-H-, and 4-CF₃-substituted aryl-bismuth reagents, respectively. Similarly, coupling with 4-ethylphenyl iodide afforded the corresponding products in 80%, 55%, and 54% yield following the same electronic trend.

While these results clearly support the role of aryl-bismuth electronics in promoting cross-selectivity, in practice the desired products could not be cleanly separated from the corresponding homocoupling products due to their very similar polarity. Accordingly, these examples are now included in the limitations section (Table S5) of the Supporting Information rather than the main scope tables. Detailed analyses quantifying the extent of homocoupling for each case have also been added to the Supporting Information (Figure S6).

Table S5. Current limitations in the substrate scope of ambiphilic cross-coupling

^aIsolated yield of the mixture.

^bYields determined by ¹H NMR analysis using dibromomethane as an internal standard

^cYields determined by ¹⁹F NMR analysis using trifluorotoluene as an internal standard.

Figure S6. Systematic analysis of side product profiles across representative substrates.

Note: In cross-couplings of aryl-bismuth reagents with aryl electrophiles, electrophile-derived homocoupling was occasionally observed (up to ~15%). In cases where the homocoupled side product exhibits similar polarity to the desired product, chromatographic separation can be challenging.

Comment 12: *Other points for consideration: There are some points where the authors should be more balanced in the description of their work. At points the discussion is exaggerated or inaccurate, which distracts from the otherwise excellent science:*

• P3: *“product 1 is efficiently obtained in 79% yield...” please remove ‘efficiently’ – at 10 mol% Pd and with 2 equivalents of ArI it is not efficient (although it is high yielding), especially in comparison to other cross-couplings (including those based on Bi nucleophiles).*

Response: The word ‘efficiently’ has been removed from this line.

Comment 13: • P4: *“and electron-donating (8) groups on the aryl iodide are well-tolerated under the reaction conditions” This is inaccurate, verging on misleading. The ethyl substituent is a very modest electron donor, and stronger EDGs fail (as shown in Table S5). The discussion should say something like ‘modest electron-donating (8) groups on the aryl iodide are well-tolerated... although stronger electron donors are not tolerated (see Table S5)’.*

Response: We thank the reviewer for their suggestion and for greater accuracy, we have amended the text include ‘moderately electron-withdrawing’ and add a note detailing that stronger electron-withdrawing groups are not as well tolerated.

Comment 14: • P5 *“heterocycles bearing acidic functionalities also afforded excellent yields”. While this is not technically incorrect, it does imply that a broader range of acidic groups is tolerated, and greater clarity would be beneficial; the authors should direct the reader to the benzyl alcohol in Table S5, which does not work.*

Response: In order to better present the strengths and limitations of the work we amended the text to say,

‘Finally, heterocycles bearing acidic functionalities also afforded excellent yields of coupling products (34, 35) without the use of an exogenous base, highlighting the robustness of the protocol, however some protic functionalities including primary alcohols still remain challenging (see Table S5).’

Comment 15: • P4: *“this protocol can tolerate substituents ortho, meta, and para to the C–I bond (5–7).” Please also draw attention to failures here with a statement like ‘for unsuccessful substrates, please see Table S5’*

Response: We have added the suggested phrase to signpost the limitations of the protocol in the main text.

Comment 16: • P5: "... piperazyl (26) groups posed no difficulties." A yield of 56% using 12.5 mol% Pd and 2 equivalents of triolborate suggests that the piperazyl group does pose difficulties. Please consider your wording here – the hyperbole just distracts from what are otherwise impressive results.

Response: We have amended the sentence to say:

'The presence of either electron-withdrawing moieties such as methyl ester (20) and trifluoromethoxy (22) groups or electron-donating methoxy (21) and piperazyl (26) groups was well tolerated.'

Comment 17: • P5: "Good yields of cross-coupled products could be obtained with aryl-bismuth species bearing electron-withdrawing or -donating groups (42–44)" cpd 44 gives 40% yield; again, please consider your use of 'good' for the yield of this compound.

Response: To better reflect the performance of the range of aryl-Bismuth reagents in the cross coupling we have reworded as follows,

'Aryl-bismuth species bearing electron-withdrawing or moderately electron-donating groups (42, 43) and heteroaryl-bismuth species bearing benzothiophene (45) and quinoline (46) aryl groups could be cross-coupled in good yields. In addition, aryl-bismuth partners bearing functionalities susceptible to oxidative addition could be employed in the reaction in moderate yield (44) (Fig. 3D).'

Comment 18: • P9 / Fig S6: the price comparisons between 'equivalent' aryl iodides/boronic acids are incorrect, and must be corrected:

o Fig 5B shows the NH pyrrole boronic acid but lists the price for the N-Boc analog. This is clearly not the same fragment as is used for the iodide, which is priced for the NH pyrrole. So either the figure must be adjusted to show the N-Boc boronic acid (and the direct comparison breaks down), or the price should be adjusted (The NH pyrrole boronic acid is available at BLD, 1 g / 786€, so the price differential is actually 1.3x more expensive for ArI, not 740x). Note that changing the figure to show the NH pyrrole boronic acid would be inappropriate because this substrate was unsuccessful, so this is not a viable ambiphilic coupling.

o choosing a 25g bottle for the Boc-pyrrole-boronic acid for the price comparison to avoid the 'price inflation' of a 1 g bottle, but the 'inflated' 1g bottle for the iodide is not appropriate. You should compare like masses to capture the same extent of 'price inflation'.

o The unfavorable comparison of the iodofuran to the furyl boronic acid is really an issue that is specific to the ambiphilic coupling. If bromides could be coupled in a general way, the price comparison would be far less dramatic (1 g / 25 USD, oakwood).

Response: We thank the reviewer for this comment and have corrected the price comparisons accordingly. The reactions shown in Fig. 5B were conducted using the N-Boc-protected pyrrole boronic acid pinacol ester, and the price analysis has been revised to consistently reflect this substrate (Figure S10).

With the corrected comparison, the corresponding aryl iodide is not commercially available, and even when compared to the NH-pyrrole iodide or the N-Boc-pyrrole bromide, these halogenated analogues are approximately 9–90-fold more expensive than the corresponding boronic derivatives. We agree that extension of the ambiphilic coupling to aryl bromides would reduce the magnitude of the price difference. Nevertheless, aryl boronic derivatives remain more broadly accessible from multiple commercial vendors, whereas electron-rich 2-haloheteroaryl halides are generally less available. A similar trend is observed for 2-halofuran substrates.

As originally noted in the manuscript, we attribute these price differences primarily to the intrinsic instability of electron-rich 2-haloheteroarenes, which leads to increased handling challenges and synthetic difficulty. Accordingly, we have revised the text to adopt a more conservative phrasing, emphasizing “*Instability-associated cost increase*” rather than overstating numerical price factors. The revised price data have been added to the Supporting Information, and the manuscript has been updated accordingly:

‘For instance, electron-rich 2-halo heteroaromatic reagents such as 2-iodopyrrole and 2-iodofuran are reported to be unstable^{37, 38}, often being less accessible and more expensive commercially, whereas the corresponding boron analogues are widely available, bench-stable, and inexpensive (see Figure S10).’

CAS	135884-31-0 (B(OH) ₂) 1072944-98-9 (Bpin)	117657-39-3	117657-37-1	67655-27-0 (I) 38480-28-3 (Br)	13331-23-2	54829-48-0	584-12-3
Merck	1 g/115 € 1g/ 10 € (Bpin) 5g/ 22 € (Bpin)	-	1 g/725 € 5 g/1455 €	-	1 g/96 € 10 g/658 €	500 mg/614 €	100 mg/44 € 5 g/150 €
TCI	1 g/54 € 5 g/170 €	-	-	-	1 g/22 € 5 g/68 €	-	1g/ 39 €
Thermo Fisher	1 g/73 € (Bpin) 5 g/281 € (Bpin)	-	-	-	1 g/55 € 5 g/169 € 25 g/640 €	-	-
BLD	1 g/10 € 1 g/10 € (Bpin) 25 g/33 € 100g/ 349 € (Bpin) 500 g/654 €	-	-	1 g/976 € (I) - (Br)	25 g/42 € 100 g/144 € 500 g/715 €	100 mg/278 €	-

*Prices are based on values as of February 10, 2026

Figure S10. Price comparison of aryl boron derivatives and electron-rich 2-haloheteroarenes from commercial vendors

Comment 19: • P9: “In certain instances, sluggish oxidative addition is commonly faced when using electron-rich aryl halides, thus making these building blocks ineffective (Fig. 4B, middle)³⁴. On the other hand, the corresponding organometallic reagent is more prone to fast transmetalation³⁴.” Some issues here: o referencing an entire text book here is not appropriate; specific primary literature should be cited to support this point (ideally relating to the actual substrates you studied).

Response: We thank the reviewer for their suggested and have now added a reference to a specific research article looking at the conducting a Hammett study on the rates of aryl iodides for the first instance the old reference was invoked: Fauvarque, J.-F., Pflüger, F. & Troupel, M *J. Organomet. Chem.* **208**, 419–427, (1981). In the second instance, we provide a reference of a Hammett study looking at transmetalation of boronic acids on Pd(II): Nishikata, T., Yamamoto, Y. & Miyaura, N. *Organometallics* **23**, 4317–4324, (2004).

Comment 20: *o Due to errors in figure and compound numbers it is not clear exactly which coupling partners you are discussing in this section, but I presumably the 3,5-dimethoxy iodide and boronic acid. The authors will be aware that the meta-OMe is electron withdrawing, not donating, so this is not a good example to use as an electron rich aryl group; please either change the example, or provide primary literature that shows this aryl group to be specifically sluggish in OA, and fast in TM. An alternative, and more compelling argument to make with this particular fragment is that the iodide cannot be easily accessed by electrophilic iodination or lithiation/iodination, whereas the boronic acid is presumably easily accessible by C-H borylation.*

Response: We thank the reviewer for this thoughtful comment. As correctly noted, *meta*-methoxy substitution does not constitute a clear electron-donating substituent. Accordingly, we have revised this section to use a *para*-methoxy-substituted aryl boronic acid (previously shown in Figure 3, compound 21), which more clearly represents an electron-rich aryl group, as the representative example.

Comment 21: *• P9: “a single aryl-bismuth scaffold can explore aryl iodides, bromides... libraries” First, this anthropomorphises the Bi reagent – “a single aryl-bismuth scaffold can be coupled with / used to explore...” is more appropriate wording. Second, the scope with respect to bromides is very limited, so ‘exploration’ of this space is not realistic. From the presented scope, a single Bi reagent could be coupled with a subset of electron-poor ArBr only. Please consider rewording to make the phrasing more representative of the reality.*

Response: In order to address these two points, we have amended the sentence to the following:

‘Taken together, these results underscore the potential of the ambiphilic cross-coupling strategy: a single aryl-bismuth scaffold can be used to engage aryl iodides, thianthrenium salts, boronic acids, pinacol boronate esters, borate libraries, and, in certain cases, aryl bromides, aryl triflates, and vinyl triflates, without the need to prepare polarity-specific substrates.’

Comment 22: *Other points for consideration:*

• P2: “This conceptually divergent framework in cross-coupling has a profound effect in the exploration of chemical space: the preparation of one aryl synthon allows the indistinctive scrutiny of both aryl nucleophiles and electrophiles at the same time (Fig. 1B).” This sentence is not clear, and does not convey what I suspect the authors actually mean – please consider revising. “conceptually divergent framework”: framework is probably not the right word here (strategy?), and it is conceptually novel, rather than conceptually divergent. “indistinctive scrutiny of both aryl nucleophiles and electrophiles at the same time” I assume that the authors mean that the Bi reagent does not distinguish between / is not limited to a single class of coupling partners, and therefore that both ArI and ArB(OH)₂ can be used. But presumably both ArI and ArB(OH)₂ cannot be used simultaneously. (this may just be my misinterpretation, but I feel that this will also confuse other readers, so is worth reconsidering to maximise the impact of the work).

Response: We thank the reviewer for their valuable input on this point. In order to better communicate the concept we have amended the sentence to read as follows:

‘This conceptually distinct cross-coupling strategy expands retrosynthetic flexibility in the synthesis of (hetero)biaryls by allowing a single aryl synthon to be employed as either a nucleophile or an electrophile (Fig. 1B). This enables the selection of coupling partners based on their synthetic accessibility or the opportunity to introduce orthogonally reactive handles for downstream functionalisation. ‘Isolable, bench stable aryl-bismuth reagents provide a modular platform for coupling reactions: in diversification of complex or high value aryl fragments they prevent repeated re-functionalisation steps in the interconversion between nucleophiles and electrophiles, thus

enabling expedient parallel synthesis across both electrophiles and nucleophiles from a single intermediate’.

Comment 23: • *Fig 1B: It would be helpful to draw the actual products of each coupling for clarity to the reader. And the meaning of “merged nucleophile & Electrophile chemical space” is not clear, please consider revising*

Response: We thank the reviewer for this helpful suggestion. In Fig. 1B, we have replaced the schematic representations with the actual structures of the coupling products to improve clarity. In addition, to better convey the intended concept, we have revised the wording ‘merged nucleophile & electrophile chemical space’ to ‘*Bridging nucleophilic and electrophilic reactivity domains*’

Comment 24: • *P3: more references are required for precedented Bi cross-couplings (seminal reports)*

Response: We thank the reviewer for this suggestion. We have added several seminal and early reports on aryl–bismuth cross-coupling between reference 17 and 18 in the revised manuscript. These include key studies by Barton, Suzuki, and Rao that established and subsequently expanded transition-metal-catalyzed C–C bond formation using aryl–bismuth reagents. The newly added references are:

18. Barton, D. H. R., Ozbalik, N., Ramesh, M. *Tetrahedron* **44**, 5661–5668 (1988).
19. Suzuki, H., Murafuji, T., Azuma, N. *J. Chem. Soc., Perkin Trans. 1*, 1593–1600 (1992).
20. Rao, M. L. N., Shimada, S., Tanaka, M. *Org. Lett.* **1**, 1271–1273 (1999).
21. Rao, M. L. N., Shimada, S., Yamazaki, O., Tanaka, M. *J. Organomet. Chem.* **659**, 117 (2002).
22. Rao, M. L. N., Venkatesh, V., Jadhav, D. N. *Tetrahedron Lett.* **47**, 6975–6978 (2006).

Comment 25: • *Fig 3A: please replace “Ar” with the full structure in each case. There is space, and it will be far easier to interpret.*

Response: We thank the reviewer for this comment. In Fig. 3A, all “Ar” labels have been replaced with the corresponding full structures to improve clarity.

Comment 26: • *P3: “aryl-bismuths”. ‘Bismuths’ is not a word; ‘aryl-bismuth reagents’?*

Response: All instances of ‘aryl-bismuths’ has been replaced by ‘aryl-bismuth reagents’.

Comment 27: • *P5: Fig 5A is called out before fig 4, so these need to be renumbered and re-ordered*

Response: We thank the reviewer for this comment. As suggested, the original Fig. 5A and Fig. 4 have been reordered and renumbered accordingly throughout the manuscript.

Comment 28: • *P7: “Importantly, these protocols do not require the use of transition metals to forge the Bi–Ar bond.” The intention of this sentence is confusing, because (to me) I was not expecting TMs to be used, and few methods for making the alternative coupling partners use TMs either (Miyaura borylation and C–H borylation are the only common methods that use TMs; electrophilic halogenation, Gignard formation, C–H lithiation do not...). Also, the use of a TM would seem preferable to the use of diazonium*

salts or organolithium reagents for Ar-Bi synthesis, so unless more context can be added please consider editing this sentence.

Response: We thank the reviewer for pointing out that our original statement regarding the absence of transition metals in the Bi–Ar bond-forming protocols could cause confusion. Our intention was to highlight that the methods used here avoid the need for transition-metal catalysts, which can be undesirable in certain contexts. However, we agree that the absence of a transition-metal step may not, in every instance, confer a general advantage, for example when the use of a transition-metal catalyst enables access to less reactive reagents. In order to soften the emphasis on the omission of transition metals in the synthesis of aryl-bismuth reagents in the main text we have changed, ‘importantly’ to ‘of note’.

Comment 29: • *Fig 5A: “...increasing chemical space exploration” I’m not sure you can claim this, but you could claim that you can increase the number of classes of coupling partners that can be engaged with a single reagent. ‘increasing chemical space exploration’ feels inappropriate because the scope in ArBr is limited (so very few of the ‘>200,000 ArBr’ can actually be used), and presumably many of the ArBr that can be used will also be represented in the ArI pool (etc), so how much is the chemical space of viable coupling partners actually increased. Please consider a clearer way of wording this point.*

Response: We have amended the caption to read: “*Ambiphilic cross-coupling from a single aryl synthon: broad coupling partner compatibility.*” This shifts the focus of the figure away from a discussion of chemical space.

Comment 30: • *Fig 4 needs ‘A’ and ‘B’ in the panels. These are discussed in the main text, but are not present in the Fig.*

Response: We thank the reviewer for pointing this out. This was inadvertently omitted, and the panel labels “A” and “B” have now been added to Figure 5 (previous Figure 4).

Comment 31: • *P9: “nucleophiles and electrophiles (Fig. 4B).” this should be Fig. 5B.*

Response: This has been implemented in the text.

Comment 32: • *In the last R&D paragraph it looks like all the compound numbers are incorrect (possibly confusion due to the incorrect figure numbers though), so I cannot comment on this discussion until referencing is corrected.*

Response: We thank the reviewer for this comment. We have carefully checked and corrected the compound numbering throughout the manuscript, including the newly added scope. All references have now been updated accordingly.

Comment 33: • *P9: “one aryl synthon is able to scrutinize both nucleophile and electrophile chemical space at the same time”*

Response: We thank the reviewer for this helpful but somewhat ambiguous comment. However, we have revised the sentence to improve clarity to read: “*By overcoming traditional polarity constraints in reagent*

design, the ambiphilic cross-coupling strategy provides pronounced synthetic flexibility, enabling a single aryl synthon to engage with both nucleophilic and electrophilic coupling partners.”

Comment 34: *Supplementary Information*

• Multiplicity should be added for ^{19}F NMR (unless acquired as $^{19}\text{F}\{^1\text{H}\}$, in which case state it for each entry), and integrals are needed where there is more than 19 F environment per molecule.

Response: We thank the reviewer for this helpful comment. In response, we have clarified that the ^{19}F NMR spectra were acquired as $^{19}\text{F}\{^1\text{H}\}$ and have corrected the notation accordingly throughout the supporting information. In addition, for molecules containing more than one ^{19}F environment, the corresponding integrations have now been explicitly reported for examples ArBi(OTf)-2 and compound **36, 49, 50, 51, 52** and **53**.

• Each known compound should be compared to literature data and a ‘matches literature data’ statement added (with a reference). This is standard.

Response: We thank the reviewer for the comment. For all known compounds, comparison with literature data has been performed and the corresponding references and “matches literature data” statements have been added.

• Technically, ^{13}C NMR chemical shifts should be reported to only 1 decimal point due to the inherent resolution of the technique (unless specific processing steps were taken).

Response: We thank the reviewer for helpful suggestion. We have carefully reviewed the Supporting Information in its entirety and revised the reported ^{13}C NMR chemical shifts to be presented consistently to one decimal place.

• Otherwise, the data are complete and the spectra beautifully clean.

Response: We sincerely thank the reviewer for this positive assessment.

We hope that the above comments and the revised manuscript are in line with the referees' suggestions and clarify your as well as the referees' requested changes. We hope that as it stands, the manuscript would be suitable for publication in *Nature*. Should you need any more information, please do not hesitate to contact me again. Thank you very much once again for handling our manuscript.

Yours sincerely,

Dr. Josep Cornella

Reviewer 1 (*referee comments in italics*; authors' replies in plain font)

This revised version of the manuscript by Cornella and co-workers includes substantial additional experimentation to address the points raised by the two reviewers. I do not have any further technical comments, and the work is fine to be published now, after which I am sure Cornella and maybe others will further study and develop the system.

With respect to the suitability of the paper for Nature, it seems the topic of discussion is whether this result (that ArBi compounds can participate both oxidative aryl transfer to Pd or redox-neutral transmetallation) is (A) a finding likely to have impact on cross-coupling practically or academically, or (B) a surprising and non-intuitive organometallic fact but without much implication on cross-coupling? In the first review, my feeling had been somewhere in the middle of the spectrum, but maybe closer to B. The result challenges the traditional electrophile/nucleophile paradigm in the same way that reductive cross-electrophile couplings did, and that is special. But it is unlikely to have comparable practical or academic impact to the cross-electrophile couplings because of how limited the results are in their current form. In that sense, this is closer to an organometallics paper than a cross-coupling paper.

Since then, I have read the other reviewer's comments and the authors' responses. Much of their analysis has merit. I am guessing many readers will agree with them, and others like me will be slightly skeptical about the impact on cross-coupling. After much reflection, I have decided that the former camp is likely to be bigger, and accordingly, I do not see any reason to hold up publication. The only way to know for sure is to come back to the paper in 5 years and see what effect it will have had. I am sure it is a landmark paper in Bi chemistry, but do not anticipate it will have similar impact in cross-coupling. It goes without saying that I would be happy to be wrong.

Response: We thank the reviewer for their positive comments and positive engagement with our response letter.

Reviewer 2 (*referee comments in italics*; authors' replies in plain font)

The authors have addressed comprehensively each of the points raised by the referees. In particular, I feel that the extra detail around side-product formation and limitations is especially valuable: as the authors will appreciate, a new method can typically only fail once in a lab before it is disregarded, so tempering expectations is important. The added detail highlights the significant practical limitations of the method (including the significant effect of electronics on cross vs homo selectivity, the potential for Bi-pincer decomposition, phosphine arylation, the lack of tolerance for ortho substitution or acidic protons, and the poor-modest performance of aryl bromides and boronic acids), but not the conceptual novelty.

The detail into the side reactions is very useful for the expert, but may be confusing to a more general user. therefore I think that a very useful addition to the SI would be a 'user guide' to help other scientists who may wish to try the chemistry, summarising the limitations and spelling out in simple terms which classes of coupling partners are best suited (eg, coupling e-poor Ar-I and e-rich Ar-Bi gives high yield and minimal homocoupling; e-poor Ar-Bi are prone to homocoupling; ArBr may benefit from Ad2PBU as ligand, and give more/less homocoupling (?); ortho substitution is generally not tolerated on either partner; etc).

One point that needs further clarification: The footnote to Figure S5 states "although the ligand-coupling product 4 was observed by GC/MS under Pd-free conditions (see Figure S15)". Figure S15 does not show the formation of product 4, or illustrate Pd-free conditions - please either revise the statement or figure accordingly..

Response: We thank the reviewer for the overall positive assessment. We agree that outlining the current limitations of the reaction in the form of a "user guide" for potential future users would be valuable. Accordingly, we have added a "user guide" to the SI at the beginning of the *General Procedure* section, where practical implementation of the reaction is described.

To clarify the Pd-free conditions, we have added the relevant information into Figure S5. The respective footnotes have been revised accordingly.

Below is an extract of what can be found in the SI:

In S38 in supplementary information

Practical summary for reaction use

A. When aryl-bismuth reagents act as nucleophiles

- a) Halide counteranions (I, Br, Cl) provide higher reactivity than non-coordinating anions.
- b) For non-coordinating anions (BF₄⁻, OTf⁻), the addition of NaI (10 equiv) improved efficiency.
- c) Homocoupling products derived from aryl iodides are commonly observed.
- d) Homocoupling products derived from aryl-bismuth reagents are occasionally observed.
- e) Electron-rich aryl-bismuth reagents (e.g., *p*-OMe) show higher reactivity.
- f) Electron-deficient aryl-bismuth reagents lead to increased homocoupling and reduced overall efficiency.
- g) *Ortho*-substitution on either aryl-bismuth or aryl-iodide decreases reactivity.
- h) Electron-rich electrophiles (e.g., *p*-OMe, *p*-NMe₂) show low reactivity.

- i) Electron-neutral aryl bromides, chlorides, and triflates are unreactive.
- j) Electron-deficient heteroaryl bromides and triflates give low yields (~5–10%) under standard conditions.
- k) Electron-deficient heteroaryl bromides, triflates, and thianthrenium salts can afford moderate yields with bulky, electron-rich ligands (e.g. PAd_2^tBu).

B. When aryl-bismuth reagents act as electrophiles

- a) Non-coordinating anions (BF_4^- , OTf^-) provide higher reactivity.
- b) Homocoupling products derived from aryl-boron reagents are generally not observed.
- c) Homocoupling products derived from aryl-bismuth reagents are occasionally observed.
- d) Electron-deficient aryl-bismuth reagents show higher reactivity.
- e) Electron-rich aryl-boron reagents show higher reactivity.
- f) Commercial aryl boronic acids and pinacol esters require the addition of K_3PO_4 or Cs_2CO_3 (2.0 equiv) to obtain moderate yields of cross-coupling.

Figure S5. Observation of aryl bismuth-derived coupling products

Note: These observations indicate that aryl-bismuth species can engage in ambiphilic coupling behavior, acting as both a nucleophilic aryl donor and an electrophilic coupling partner. Consistent with this interpretation, aryl bismuth-derived coupling products (3 and 4) are observed in the absence of aryl iodide, supporting the possibility of off-cycle reactivity (see Figure S15).

We hope that the above comments and the revised manuscript are in line with the referees' suggestions and clarify your as well as the referees' requested changes. We hope that as it stands, the manuscript would be suitable for publication in *Nature*. Should you need any more information, please do not hesitate to contact me again. Thank you very much once again for handling our manuscript.

Yours sincerely,

Dr. Josep Cornella

Director Max-Planck-Institut für Kohlenforschung